# Tumor Spheroids as Model to Design Acoustically Mediated Drug Therapies: A Review

**DOI:** 10.3390/pharmaceutics15030806

**Published:** 2023-03-01

**Authors:** Marie Roy, Corentin Alix, Ayache Bouakaz, Sophie Serrière, Jean-Michel Escoffre

**Affiliations:** 1UMR 1253, iBrain, Université de Tours, Inserm, 37032 Tours, France; 2Département d’Imagerie Préclinique, Plateforme Scientifique et Technique Analyse des Systèmes Biologiques, Université de Tours, 37032 Tours, France

**Keywords:** multicellular tumor spheroid, drug delivery, microbubble-assisted ultrasound, sonodynamic therapy, therapeutic ultrasound, mild hyperthermia, cavitation

## Abstract

Tumor spheroids as well as multicellular tumor spheroids (MCTSs) are promising 3D in vitro tumor models for drug screening, drug design, drug targeting, drug toxicity, and validation of drug delivery methods. These models partly reflect the tridimensional architecture of tumors, their heterogeneity and their microenvironment, which can alter the intratumoral biodistribution, pharmacokinetics, and pharmacodynamics of drugs. The present review first focuses on current spheroid formation methods and then on in vitro investigations exploiting spheroids and MCTS for designing and validating acoustically mediated drug therapies. We discuss the limitations of the current studies and future perspectives. Various spheroid formation methods enable the easy and reproducible generation of spheroids and MCTSs. The development and assessment of acoustically mediated drug therapies have been mainly demonstrated in spheroids made up of tumor cells only. Despite the promising results obtained with these spheroids, the successful evaluation of these therapies will need to be addressed in more relevant 3D vascular MCTS models using MCTS-on-chip platforms. These MTCSs will be generated from patient-derived cancer cells and nontumor cells, such as fibroblasts, adipocytes, and immune cells.

## 1. Introduction

The active and targeted delivery of anticancer drugs (e.g., chemotherapeutics, therapeutic antibodies, tyrosine kinase inhibitors, nucleic acids, etc.) into tumor tissues is a major challenge in oncology to eradicate tumors and to avoid their recurrence and the formation of metastases while reducing the side effects of these treatments. In this context and for decades, drug delivery methods have been designed and first validated using in vitro models. Two-dimensional in vitro tumor models, including monolayer cell culture, are mostly used for drug screening, drug design, drug targeting, and drug toxicity and for the validation of drug delivery methods. However, several of these pharmaceutical developments, which have shown great promise in vitro, failed in vivo. The main explanation for such a failure lies in the fact that two-dimensional (2D) in vitro tumor models do not reflect the three-dimensional (3D) organization of a tumor in vivo and its pathophysiology [1]. In these 2D models, tumor cells are more sensitive to anticancer drugs than in vivo because these drugs have direct access to cell membranes and specifically to membrane receptors, targets of drugs, or targeted-drug-loaded carriers. In addition, these tumor cells are clones that have the same physiology and are grown in monoculture. Such 2D models do not reflect the in vivo context and specifically the influence of the tumor microenvironment on drug resistance and efficacy [2]. Indeed, the extracellular matrix (a noncellular meshwork of crosslinked collagens, proteoglycans, and glycoproteins) operates in coordinated mode with tumor cells and nontumor cells such as cancer-associated fibroblasts, cancer-associated adipocytes, and tumor-infiltrating lymphocytes. This meshwork plays a major role in the 3D architecture of tumors and their tumor microenvironment thus facilitating the interaction between the different kinds of cells through tight intercellular junctions. All these cells communicate with each other through the production and secretion of signaling molecules such as chemokines, cytokines, hormones, and growth factors [3]. The extracellular matrix also notably affects the biodistribution of drugs and their access to tumor cells. In addition, abnormal vascularization observed in the core of the tumor does not allow a supply of nutrients and oxygen, which generates hypoxic regions resistant to anticancer treatments, particularly radiotherapy but also chemotherapy [4]. Altogether, these molecular and cellular actors, as well as their interactions with each other, create a complex environment that influences the intratumoral biodistribution, pharmacokinetics, and pharmacodynamics of drugs.

Accordingly, animal models are still essential to investigate these pharmacological properties. However, since 1959, the 3Rs principle (Reduce, Refine, Replace) has provided a stricter framework in animal experimentation. Two of these 3Rs aim to reduce the number of animals and to replace their use with alternative methods as soon as possible. In this context, 3D in vitro tumor models have been developed to overcome the limitations of the 2D in vitro tumor models described above and to better predict the in vivo response to anticancer therapies while limiting the use of animal models [5]. Among these 3D models, tumor spheroids or multicellular tumor spheroids (MCTSs) are the most commonly used models for the design and validation of chemotherapeutic strategies. These spheroids are 3D clusters of tumor cells without (termed spheroids) or with (termed MCTS) other types of cells (e.g., fibroblasts, adipocytes, immune cells, etc.). Depending on their cell composition and size, the 3D organization of spheroids creates a gradient of oxygen, nutrients, and metabolites from the periphery to the core of the spheroids, thereby generating an inverse gradient of carbon dioxide and metabolic waste [6,7]. Both of these gradients are responsible for forming a necrotic core, an intermediate layer of quiescent cells, and an outer layer of proliferative cells, thus partially mimicking in vivo tumor physiology [8]. Indeed, their architectures give the spheroids greater resistance to drugs than that observed in 2D in vitro models, thus improving the predictive tumor drug sensitivity [9,10]. In addition, the interaction of a dense extracellular matrix with the spheroid’s cells (which themselves interact with each other via intracellular junctions) creates a molecular and cellular network that simulates, albeit partly, the physiological barriers encountered in vivo during the evaluation of drug penetration into the tumors [11,12,13]. MCTSs are composed of tumor cells and other types of cells (e.g., fibroblasts, adipocytes, immune cells, etc.) that can also more precisely but still partially mimic tumor heterogeneity and its microenvironment, which influence the efficacy of drug delivery and the therapeutic benefit of the treatment [14]. These MCTSs also make it possible to investigate the molecular targeting of anticancer therapies [15]. Regarding the advantages previously stated about the physiological properties of spheroids, these 3D models are now widely used in pharmaceutical oncology to develop innovative treatments for cancer.

In oncology, the efficacy of systemic chemotherapy is often far from sufficient because of the tumor microenvironment, which restricts both intratumoral drug bioavailability and the targeting of tumor cells. In addition, these systemic chemotherapies are frequently related to severe off-target effects as a consequence of the nonspecificity of these therapies. In this regard, the development and validation of targeted drug delivery methods are required to increase the therapeutic dose of the drug into the tumor tissue while minimizing side effects to healthy tissues. These drug delivery methods might have a great benefit in the design of first-line, neoadjuvant, and palliative care. Among these methods, ultrasound (US) is a promising delivery method for several therapeutic molecules (e.g., chemotherapeutic drugs, immunotherapeutic agents, nucleic acids, and sensitizing agents) in various kinds of tumors. Depending on acoustic parameters, ultrasound can generate both thermal and mechanical stimuli, which may be exploited (i) to trigger the release of drugs from particles (e.g., liposomes, polymer particles, micelles, microbubbles, etc.), (ii) to excite sensitizers, and (iii) to induce the reversible permeabilization of biological barriers (e.g., endothelial barriers, plasma membrane, etc.), thus improving the accumulation of drugs in tumor cells [16,17,18,19,20,21,22,23,24,25]. These US strategies, including mild hyperthermia, sonodynamic therapy, microbubble-assisted US, and US on its own, improve the efficacy of anticancer therapies in a safe and noninvasive way [26,27,28,29]. In addition, these cost-effective strategies are applicable to a wide range of superficial and deep tumors. For a few decades, drug-loaded particles and drug delivery protocols using therapeutic US have been evaluated in spheroids [30,31,32,33]. In the present review, we first provide a survey of different spheroid culture system methods and then of in vitro investigations, which exploit spheroids to design acoustically mediated drug delivery methods and to evaluate their therapeutic efficacy. The limitations and future perspectives of these studies are also debated.

## 2. Methods

The PubMed^®^ and Web of Science^TM^ electronic databases were screened by two authors of this review (M.R. and J.-M.E.) using predefined search dates (January 1991–July 2022) and terms for the in vitro 3D spheroid model and therapeutic US for anticancer drug delivery. The PubMed^®^ and Web of Science^TM^ database search terms used were ((sonodynamic therapy) AND (spheroid)) OR ((ultrasound) AND (spheroid) AND (drug delivery)) OR ((ultrasound-induced cavitation) AND (spheroid)). An “English language” filter was applied. Table 1 summarizes the inclusion and exclusion criteria. The selection process is outlined in Figure 1. Twenty-four publications met our inclusion criteria. These publications were sorted by the type of therapeutic US used: 11 publications with sonodynamic therapy, 7 publications with microbubble-assisted US, 2 publications with US-mediated mild hyperthermia, 2 publications with low-intensity pulsed US and gold nanoparticles, 1 publication with acoustic cavitation, and 1 publication with laser-generated focused US.

## 3. Results

### 3.1. Different Spheroid Culture System Methods

In the present section, the different spheroid culture system methods from the simplest to implement, such as the hanging drop method [34], liquid overlay technique or nonadherent culturing surface [35,36], and magnetic levitation [30,37], to the most complex, including matrix- or scaffold-based methods [38,39,40,41,42,43,44,45,46], microfluidic devices [47,48], and bioreactor culture systems for large-scale production, are described. The choice of a spheroid culture system method depends not only on the type and quantity of spheroid required but also on the technical and financial means available [36,49]. The advantages and limitations of these main methods are shown in Figure 2.

#### 3.1.1. Hanging Drop Method

The hanging drop method is the simplest and least expensive technique to implement, but it does not allow obtainment of a large number of spheroids. This method consists of depositing a drop of cell suspension in the lid of a Petri dish. The lid is then turned over, and the drop remains suspended due to surface tension [34]. Gravity induces the aggregation of cells, thus facilitating the formation of spheroids. A saline solution is added to the bottom of the Petri dish to prevent evaporation of the drops. However, changing the culture medium in this configuration remains difficult and time-consuming, resulting in limited production of spheroids. Currently, more elaborate multiwell cell culture plates have been developed to replace Petri dishes [50]. The tops of these plates have small holes that allow drops to be placed in a simple way, thus resulting in a higher yield and more homogeneous spheroids.

#### 3.1.2. Forced Floating Methods

Forced floating methods are partly based on the use of plastic surfaces treated to prevent the attachment of cells to these surfaces and to promote their aggregation. Thus, so-called “ultralow attachment” (ULA) multiwell microplates are marketed. The tumor cells are seeded and centrifuged. Then, spheroids are formed after a few days [51]. Among forced floating methods, the “liquid overlay” technique involves adding an inert liquid to the bottom of the culture well, such as agarose or poly(2-hydroxyethyl methacrylate) (pHEMA), to limit cell adhesion [35,51]. All these methods are inexpensive, easy to set up, and reproducible.

#### 3.1.3. Rotary Cell Culture/Agitation-Based Culture

The agitation-based culture method relies on the culture of tumor cells under permanent agitation, which prevents sedimentation and promotes cell–cell interactions, thus resulting in the formation of spheroids. This method provides a large-scale production of spheroids. Several systems are commercially available, such as the “spinner flask”, where a rod with two paddles agitates the cell culture medium continuously inside the flask [52]. In this system, the agitation parameters are crucial to avoid excessive shear forces that could affect cell physiology. If this system makes it easy to supply nutrients and to eliminate cell waste, it consumes a large amount of cell culture medium. Another limitation of this device is the heterogeneous size of the spheroid obtained, which consequently requires manual screening of the latter to obtain a population of homogenous spheroids. To homogenize the size of the spheroid, it is also possible to combine techniques, for example, by initiating the formation of spheroids in a liquid overlay and then transferring them to agitation-based culture systems for their growth [53].

Based on the same principle, other devices are grouped under the term “rotating cell culture bioreactors” (e.g., roller tube, rotating wall vessel), in which a tube containing a suspension of cells rotates on itself, providing constant agitation. For example, the rotating wall vessel system consists of a cylinder containing a cell culture chamber surrounding a rotator, which allows the rotation of this chamber along a horizontal axis at constant speed. Thus, the gravitational force is counterbalanced with a hydrodynamic force that maintains cells in suspension by creating a microgravity that decreases the shear stresses compared to the spinner flask while allowing the homogeneous circulation of nutrients and O_2_ in the medium [54].

#### 3.1.4. Matrix-Based Methods

The use of semisynthetic [46], synthetic (poly(lactic-co-glycolic) acid, polycaprolactone, poly(ethylene glycol)), and biological (Matrigel^®^ [40], gelatin [41], alginate [42,43], collagen [44,45]) matrices can induce the formation of spheroids and reproduce the interaction between the tumor cells and the extracellular matrix met in vivo. These cells are grown on the matrix or are included in the matrix [38,39,53]. Among the matrices, the most widely used is Matrigel, a solubilized basement membrane matrix secreted by Engelbreth-Holm-Swarm mouse sarcoma cells containing laminin, collagen, elastin, entactin, fibronectin, and fibrinogen and supplemented with numerous growth factors [40]. Once gelled, Matrigel forms a mesh with viscoelastic properties comparable to those of the extracellular matrix in vivo. Then, the tumor cells organize themselves in 3D and interact with the different constituents of the Matrigel [46]. The main limitation of these biological matrices is their composition, which can vary between different batches, thus resulting in a lack of reproducibility in the studies. Indeed, spheroids of different sizes can be obtained, making pharmaceutical studies difficult. To overcome these issues, semisynthetic or synthetic hydrogels have been designed. The composition and properties of these hydrogels are controlled, resulting in more reproducible results. Even if these hydrogels are biologically inert and not biodegradable, RGD peptide units for β-integrin-mediated cell adhesion as well as sites biodegradable through metalloproteinases can be included in the hydrogels. The development and purification of such hydrogels remain complex and expensive. Therefore, hybrid hydrogels combining the advantages of biological and synthetic hydrogels have been developed. These hydrogels include natural constituents that promote adhesion and reproduce the viscoelastic properties of the extracellular matrix in vivo [55].

#### 3.1.5. Magnetic Levitation or Printing

Magnetic levitation relies on the incubation of paramagnetic iron oxide nanoparticles with tumor cells overnight. As a result, these nanoparticles are internalized by the cells. After removing excess nanoparticles, the cells are seeded in ULA plates. A magnet is then placed on top of the plate to isolate the cells and to promote their aggregation. It is also possible to place the magnet underneath the plate, and in this case, the process is called “magnetic printing”. At the recommended concentration, nanoparticles do not induce toxicity, but the equipment needed, such as some commercially available nanoparticles or magnet plates, can be relatively expensive [56,57]. These techniques have several advantages, including the possibility of maintaining spheroids in culture for long periods of time (beyond 12 weeks) and the fact that spheroid formation can be very rapid within a few hours [58,59,60]. The magnetic forces between the magnet and the spheroids can also be extremely useful when changing media or manipulating the spheroids, facilitating maintenance and limiting the loss of spheroids [57].

#### 3.1.6. Microfluidic Systems

Microfluidic systems can be useful either for spheroid production or for the maintenance of spheroids in culture under controlled fluid conditions [47]. Spheroid formation using microfluidic systems can be classified into two main categories: emulsion-based methods for microdroplet formation generating spheroids encapsulated in a matrix or microarray-based methods allowing spheroid formation by capturing cells in microwells [47]. On the other hand, the commercially available microfluidic systems for spheroid culture make it possible to reproduce the biological flows encountered in vivo to ensure the supply of nutrients to the spheroid and to evacuate metabolic waste [47]. Each system has its own way of working, but in general, they are made up of an inlet and an outlet connection between which, via microfluidic chains, the liquid will circulate toward the microwells containing the spheroid [48,61]. The flow parameters and the composition of the cell culture medium as well as that of the extracellular matrix can be precisely controlled using these systems [59,61].

### 3.2. Contribution of MCTS to the Design of Acoustically Mediated Drug Therapies

In this section, we provide a review of in vitro investigations that exploit spheroids to design and evaluate acoustically mediated drug therapies. First, the principle of each US method on which these therapies are based will be introduced. Then, we will explain how the use of spheroids allowed the development of these drug therapies.

#### 3.2.1. Drug Delivery Using Microbubble-Assisted Ultrasound

Microbubble-assisted ultrasound (MB-assisted US) is a noninvasive and targeted drug delivery method that enhances the therapeutic efficacy of anticancer drugs (e.g., chemotherapeutic drugs, oncolytic viruses, kinase inhibitors, nucleic acids, immunotherapeutics, and sonosensitizers) by increasing their intratumoral biodistribution and reducing their off-target effects [62,63,64,65,66]. In vivo, these drugs are either co-injected or injected sequentially with MBs or loaded on or into the MBs before their administration [67]. After a sufficient accumulation of drugs and MBs into the tumor tissue (after intratumoral injection) or into the tumor microvasculature (after intravenous injection), the tumor tissue is exposed to US. Subsequently, the US-mediated volumetric oscillations of MBs generate a number of local acoustic events (e.g., pulling/pushing process, microstreaming, shock waves, and microjet) near the plasma membrane of tumor cells or the blood–tumor barrier, which promote their reversible permeabilization. This enhances the extravasation, penetration, and retention of anticancer drugs into tumor tissues through the stimulation of paracellular and transcellular pathways. The efficacy and safety of this acoustically mediated drug delivery method strongly depend on (i) the pharmacological properties of anticancer drugs and MBs, (ii) the physiology of tumor tissue, (iii) US devices and parameters, and (iv) treatment schemes. Since the advent of spheroids, the latter are increasingly used to investigate the efficacy of US protocols and new formulations of drugs or/and MBs, or to evaluate the influence of the tumor microenvironment on the efficacy of treatment (Table 2).
Evaluation of proof of concepts

Doxorubicin (Dox) is a potent anticancer drug prescribed on its own or in combination with other drugs for the treatment of solid tumors. However, its clinical use is still rather limited because of its off-target effects. To overcome this issue, Dox is either actively administered in free form using drug delivery methods or encapsulated inside pegylated liposomes (Doxil^®^ or Caelyx^®^) or polymeric nanoparticles, which can be passively administered but also actively delivered. Thus, Paškevičiūtė et al. [68] evaluated the influence of MB-assisted US on the delivery of doxorubicin (Dox) into monolayer cells and MCTS of triple-negative breast cancer (MDA-MB-231) or non-small-cell lung cancer (NSCLC) (A549) cells cocultured with human fibroblasts. MCTSs were formed using a magnetic 3D bioprinting method. Then, cell monolayers and MCTSs (200–225 μm) were exposed to MB-assisted US (1 MHz, 10 Hz pulse repetition frequency (PRF), 50% duty cycle (DC), 0.5 MPa peak negative pressure (PNP) for 20 s or 2 min; SonoVue^®^ MBs: 20 μL) in the presence of Dox (10 μM). Using the native fluorescence of Dox, the penetration and distribution of this drug into the MCTS were monitored by fluorescence microscopy. Regardless of the US exposure time, MB-assisted US did not increase the intracellular Dox fluorescence in cell monolayers compared to Dox treatment alone. Nevertheless, the cytotoxicity of Dox after MB-assisted US was enhanced by approximately 5.7% in comparison with Dox treatment alone. In addition, the exposure of NSCLC MCTSs to MB-assisted US for 20 s induced 1.4-fold and 1.8-fold increases in Dox fluorescence into the edge and middle regions of the MCTS 1 h post-US exposure, respectively. However, a two-fold increase in Dox fluorescence was detected only in the middle regions of spheroids 2 h post-US exposure. A US exposure time of 2 min did not enhance Dox fluorescence in either region of breast cancer MCTSs 1 h post-US exposure, while there was a 1.6-fold increase in Dox fluorescence in both regions 2 h after US exposure. Although these data show that MB-assisted US increases the penetration and accumulation of Dox in both NSCLC and breast cancer MCTSs, we regret that the authors did not seek to correlate this enhanced bioavailability of Dox with an increase in its cytotoxicity, by investigating its effect on MCTS viability and growth.

**Table 2 pharmaceutics-15-00806-t002:** Drug delivery using MB-assisted US.

Ref.	Drug, Dye, Particles	Cell Line	Spheroid Formation Method	US Parameters	Microbubbles	Drug Distribution	Cytotoxicity Assay	Main Outcomes
[69]	Dox (543.52 Da)	MDA-MB-231 (BC cell line)	ULA plate with Geltrex matrix	1 MHz, 8 cycles, 2 ms PRP, for 30 s	Definity^TM^	Confocal microscopy	Live/dead cell viability assay Monitoring of spheroid size	Higher diffusion coefficient of Dox in deep region Significant reduction in spheroid growth
[68]	Dox	MDA-MB-231 - A549 (NSCLC)	Magnetic 3D bioprinting - Coculture with a ratio 1:1	1 MHz, 50% DC, 0.5 MPa for 20 s or 2 min	SonoVue^®^	Confocal microscopy	NA	Increase in Dox fluorescence into the middle layer and not in the central region of BC and NSCLC MCTS
[70]	Fluorescent polystyrene beads Fluorescent labeled liposomes (Dox)-loaded thermosensitive liposomes carried on MBs (100–200 nm)	NIH/3T3 (Fibroblasts) - 4T1 (BC cell line)	Microwell array chip coat with agarose mono- and coculture	1 MHz, 10% DC, 2000 cycles/pulse, 2 W cm^−2^ for 10 s	Lab-made lipid-shelled MBs	Flow cytometry Confocal microscopy	CellTiter-Glo^®^ cell viability assay Incucyte live cell analysis Luciferase assay	Efficient delivery of liposomes and Dox in the outer layer of the spheroid Increase in the intracellular uptake of liposomes and Dox Significant increase in Dox delivery in spheroids compared to MCTSs MCTSs are less sensitive to the treatment compared to spheroids Acoustically mediated Dox delivery using (Dox)-loaded thermosensitive liposomes carried on MBs is more efficient than co-administration approach
[71]	Gem (263.20 Da) PTX (853.90 Da)	Panc-01 (PDAC cancer cell)	Liquid overlay: Plate coated with agarose	1 MHz, 100 Hz PRF, 30% DC, 0.48 MPa PNP for 30s	Lab-made drug-loaded, lipid-shelled MBs	NA	Spheroid morphology MTT assay	The combination of Gem/PTX-loaded MBs with US induced higher cytotoxic effects than Gem-loaded MBs with US
[31]	Ox (397.29 Da) Ir (586.70 Da)	Panc-01	Liquid overlay: Plate coated with agarose	1 MHz, 100 Hz PRF, 50% DC, 3 W cm^−2^ for 30 s	Lab-made drug-loaded, lipid-shelled MBs	NA	Spheroid morphology Propidium iodide staining	Acoustically mediated Ir delivery using Ir-loaded MBs significantly decreased the viability of spheroid than the acoustically mediated Ir delivery without MBs The combination of Ox/Ir-loaded MBs with US induced significant inhibition of spheroid growth compared to those treated with Ir-loaded MBs and US
[48]	Dox	HCT116 (CC cell line) - HFFF2 (Human fetal foreskin fibroblast)	ULA plate Microfluidic system - Coculture with a ratio 1:1	1 kHz PRF, 1% DC, 0.81 MPa PNP, for 2 s	Lab-made lipid-shelled MBs	Confocal microscopy	Spheroid morphology CellTiter-Glo^®^ Cell Viability Assay NucRed Dead labelling	Dox delivery using MB-assisted US induced a significant increase in Dox uptake into the spheroid and a significant decrease in cell viability Acoustically mediated Dox delivery using Dox-loaded MBs is more efficient in Dox uptake and cell mortality than the co-delivery of Dox-loaded liposomes with MBs +US or Dox-loaded liposomes alone MCTS are most resistant to Dox (IC50 = 1.9 µM) compared to spheroids (IC50 = 0.9 µM)
[72]	Fluorescent cationic, anionic and neutral nanoparticles (20, 40, 100 nm)	MCF-7 (BC cell line)	Agitation based method with agarose beads	1 MHz, 10 Hz PRF, 10–50% DC, 0.5 MPa for 10–90 s	Optison^®^	Confocal microscopy	NA	Penetration into spheroids decreased when nanoparticle size increased Anionic nanoparticles penetrated deeper than neutral and cationic nanoparticles Nanoparticles penetration also depended on time exposure and duty cycle

Dox = doxorubicin; BC = breast cancer; ULA = ultralow attachment plate; PRP = pulse repetition period; NSCLC = nonsmall cell lung cancer; DC = duty cycle; NA = not available; MBs = microbubbles; MCTS = multicellular tumor spheroid; Gem = gemcitabine; PTX = paclitaxel; PDAC = pancreatic ductal adenocarcinoma; PNP = peak negative pressure; PRF = pulse repetition frequency; Ox = oxaliplatin; Ir = irinotecan; CC = colon cancer IC50 = half maximal inhibitory concentration.

Moreover, Misra et al. [69] investigated the influence of MB-assisted US on the penetration, distribution, and efficacy of doxorubicin (Dox) in a breast cancer (MDA-MB-231) spheroid model. To achieve this objective, spheroids were generated by seeding and by growing breast cancer cells in a ULA microplate using Geltrex as the extracellular matrix. These spheroids (size 300–400 μm) were incubated with Dox (50 μM) and MBs (Definity^TM^ at 1.7% *v*/*v* concentration) under magnetic steering. They were then exposed to US waves (1 MHz, 8 cycles, 2 ms pulse repetition period (PRP), 770 kPa PNP for 30 s). The penetration and accumulation of Dox into the spheroid were monitored by confocal laser scanning microscopy (CLSM). The results revealed that MB-assisted US significantly increased the penetration of Dox into the deeper regions of spheroids. After spheroid dissociation, the intracellular uptake of Dox was assessed using flow cytometry. This analysis confirmed that acoustically mediated Dox delivery significantly enhanced Dox uptake per cell by 25% and was associated with a 1.2-fold increase in total fluorescence compared to Dox treatment alone. The therapeutic efficacy of this treatment was assessed 4 h after Dox delivery using a live/dead cell viability assay (Cellbrite^TM^ Orange/Nuclear Blue^TM^ DCS1) with flow cytometry. The analysis of this early cell mortality did not show a significant increase in cell death after acoustically mediated Dox delivery compared to Dox treatment alone, despite a significant increase in the penetration and the distribution of Dox into the spheroid. Accordingly, the authors analyzed the therapeutic efficacy of this treatment over a long-term period by monitoring the growth of the spheroids for 28 days after treatment. As reported in the short-term period (4 h after treatment), no significant difference was observed in spheroid size after acoustically mediated Dox delivery compared to Dox treatment alone. Altogether, these results show that MB-assisted US increases the penetration and intracellular uptake of Dox into deeper spheroid layers. However, surprisingly, this improvement does not translate into an increase in the therapeutic efficacy of Dox, suggesting that the increased dose of Dox into the spheroid is not enough to induce an additional loss of cell viability in comparison with Dox treatment alone. Nevertheless, we think that an optimization of US- and MB-related parameters should make it possible to establish a positive correlation between a significant accumulation of Dox in the spheroids and a significant decrease in their growth.
Influence of acoustic parameters on dye/nanoparticle delivery

Recently, our research group investigated the influence of peak negative pressure (PNP) on the delivery of the fluorescent dye, propidium iodide (PI) (used as a drug model), into colorectal cancer spheroids [73]. The colorectal cancer cells (HCT-116) were seeded and cultured in a ULA 96-well microplate. The spheroids (300 μm in diameter) were incubated with PI (100 μM) and Vevo MicroMarker^TM^ MBs (30 μL; 2 × 10^9^ MBs/mL). Then, they were immediately exposed to US waves (1 MHz, 100 μs BP, 40% DC, 100 kPa to 400 kPa) for 30 s. The dye delivery into spheroids was assessed by measuring the fluorescence intensity of PI with fluorescence imaging microscopy. As shown in Figure 3A,B, the exposure of spheroids to an acoustic pressure of 100 kPa in the presence of MBs significantly induced an increase in the fluorescence intensity into the spheroids compared to PI treatment alone (*p* < 0.05), suggesting a significant enhancement of the penetration and the accumulation of PI into the spheroids. The increase in the acoustic pressure from 200 kPa to 300 kPa resulted in an additional increase in the fluorescence intensity compared with that obtained at 100 kPa (*p* < 0.05). At both 300 and 400 kPa, a similar fluorescence intensity was detected in spheroids (*p* > 0.05). As displayed in Figure 3B, the increase in the acoustic pressure from 100 to 300 kPa induced a linear increase in the fluorescence intensity before reaching a plateau. The influence of acoustic pressure on spheroid growth and viability was determined by measuring the spheroid area for 13 days and by using trypan blue staining on the 13th day, respectively. As shown in Figure 3C,D, the increase in acoustic pressure from 100 to 400 kPa did not affect spheroid growth and viability. Altogether, these results suggest that acoustic pressure is a key parameter in improving the delivery of small molecules into spheroids.

Moreover, Grainger et al. [72] investigated the effects of the size (20, 40, and 100 nm) and surface charge (anionic, cationic, and neutral) of fluorescent nanoparticles (NPs) as well as US parameters (duty cycle, US exposure time) on the penetration of these NPs into breast cancer spheroids. Breast cancer cells (MCF-7) were seeded on agarose beads and cultured in an orbital shaker to generate spheroids. These spheroids (300–350 μm) were exposed to US waves (1 MHz, 10 Hz PRF, 10–50% DC, 0.5 MPa for 10–90 s) in the presence of 3.64 × 10^13^ fluorescent NPs and MBs (Optison^®^—50 μL). The adhesion and penetration of NPs on or inside the spheroids were assessed using optical imaging microscopy. The simple incubation of spheroids with 20 and 40 nm NPs induced high adhesion and penetration into the spheroid surface and intermediate layer compared to deeper layers. Regardless of the US parameters, the exposure of spheroids to MB-assisted US increased the penetration of NPs into the spheroid core in a size-dependent manner. Nevertheless, the acoustically enhanced penetration of NPs into spheroids strongly depended on DC. Thus, 30% DC was the optimal DC to induce the highest penetration and retention of 20 nm NPs into the spheroids without affecting the spheroid integrity and viability. In addition, the concentration of 20 nm anionic NPs in the core, intermediate layers and surface of the spheroid was nearly twice as much as that of neutral and cationic NPs. Anionic NPs should more easily penetrate through the spheroid’s interstitium because of the electrostatic repulsion with the negatively charged patterns present at the surface of tumor cells and in the extracellular matrix embedding these cells. These results show that the acoustically mediated delivery of small anionic nanoparticles is a suitable strategy to significantly accumulate drugs into the core of tumors.
Evaluation of the new formulation of drug-loaded MBs

Gao et al. [31] designed a new formulation of liposome-loaded MBs carrying irinotecan (Ir) and oxaliplatin (Ox). Both chemotherapeutic drugs are commonly associated with 5-fluorouracil and folinic acid for the treatment of advanced pancreatic cancer. This chemotherapeutic regimen, also called FOLFIRINOX, has shown a great benefit on patient survival, but this regimen is only prescribed for patients with good physical condition because of its acute side effects. In this new formulation, oxaliplatin was encapsulated into liposomes, which were then conjugated on lab-made and phospholipid-shelled MBs containing irinotecan (Ir/Ox-loaded MBs). Pancreatic tumor cells (Panc-01) were seeded and cultured on 96-well plates coated with agarose solution to generate spheroids 96 h after seeding. These spheroids were incubated with Ir-loaded MBs (50 μM Ir and 9.55 × 108 MB/mL) or Ir/Ox-loaded MBs (low dose: 25 μM Ir and 129 μM Ox; high dose: 50 μM Ir and 258 μM Ox) and then exposed to US waves (1 MHz, 100 Hz PRF, 50% DC, 3 W cm^−2^) for 30 s. The efficacy of treatment was measured 48 h later by observing the size and integrity of spheroids using optical imaging and by staining the spheroid with PI, a fluorescent dye that labels the dead cells. First, the acoustically mediated irinotecan delivery using Ir-loaded MBs into spheroids induced a 42% increase in the PI fluorescence compared to those treated with free irinotecan delivered with US alone. As a result, the former spheroids displayed a loss of spheroid cohesion and a smaller size than the latter ones. In addition, the exposure of spheroids to MB-assisted US with a low dose of Ir/Ox-loaded MBs resulted in a substantial decrease in spheroid size and loss of spheroid integrity, while the PI fluorescence was not significantly increased compared to the spheroids treated with a low dose of Ir/Ox-loaded MBs without US exposure. The spheroids incubated only with a high dose of Ir/Ox-loaded MBs (i.e., without US application) were also smaller with reduced spheroid cohesion and showed higher PI fluorescence (32%) than those exposed to a low dose of Ir/Ox-loaded MBs in the presence of US. The acoustically mediated drug delivery into the spheroids using a high dose of Ir/Ox-loaded MBs induced a further 52% in PI fluorescence compared to those treated only with this MB dose. This fluorescence was significantly higher than that of spheroids exposed to US in the presence of Ir-loaded MBs. These results indicate that MB-assisted US using Ir/Ox-loaded MBs is more effective than MB-assisted US using Ir-loaded MBs in this spheroid of pancreatic cancer. To confirm these results in vivo, the Ir/Ox-loaded MBs (4.75 ± 0.83 mg/kg Ir; 0.91 ± 0.34 mg/kg Ox) were administered intravenously in a subcutaneous pancreatic cancer (BxPC3) xenograft tumor model. Once tumors reached approximately 100 mm^3^, they were exposed to US (1 MHz, 30% DC, 100 Hz PRF, 0.48 MPa PNP) for 3.5 min. The tumor dimensions were measured using a Vernier caliper over time. This treatment resulted in the inhibition of tumor growth compared to control conditions (no treatment, free Ox/Ir or Ir/Ox-loaded MBs). If these in vivo results surprisingly confirmed the results obtained with the spheroids, we can see here two limitations: (i) the tumor cell lines used for the spheroids and in vivo studies are different (Panc-01 versus BxPC3), and it is unlikely that these cell lines have the same sensitivity to the treatment; and (ii) the use of the avascular spheroids. The spheroids were incubated directly with the Ir/Ox-loaded MBs whereas they were administered intravenously in vivo. Indeed, the spheroids were incubated directly with the free drugs or drug-loaded MBs whereas these latter were administered intravenously in vivo. In the spheroid model, MB-assisted US induces (i) the release of the drugs from MBs, (ii) the loss of spheroid cohesion thus facilitating the drug penetration inside the spheroids, and (iii) the permeabilization of the tumor cells thus improving the intracellular uptake of drugs. In the in vivo tumor model, MB-assisted US may permeabilize the tumor microvasculature and consequently enhance the drug extravasation and its intratumoral bioavailability. This permeabilization may increase the intracellular delivery of drugs in the endothelial cells, thus potentiating the destruction of tumor vasculature and reducing the nutrient supply of tumors. It is well known that because the MBs are pure vascular agents, MB-assisted US has no direct effect on tumor cells in vivo compared to spheroid models. Nevertheless, this study is the first to report on the acoustically mediated co-delivery of two chemotherapeutic drugs using liposome-loaded MBs.

In the same way, Logan et al. [71] generated stable MB formulations loaded with gemcitabine (Gem-loaded MBs) or a combination of gemcitabine and paclitaxel (Gem/PTX-loaded MBs). This drug combination is a standard of care chemotherapy regimen used in the treatment of patients with advanced pancreatic cancer. It has shown a survival benefit compared to gemcitabine monotherapy. Nevertheless, this regimen is associated with severe off-target toxicity. Both drugs were incorporated into the MB shells during the self-assembly process. As previously described, a suspension of Panc-01 cells was seeded and cultured on microplates coated with agarose solution for 4 days to establish spheroids. These spheroids were incubated with Gem-loaded MBs or Gem/PTX-loaded MBs (10 μM Gem; 6.2 μM PTX; 3 × 10^7^ MBs) and then immediately exposed to US waves (1 MHz, 100 Hz PRF, 30% DC, 0.48 MPa) for 30 s. Two days later, spheroid morphology was observed using optical microscopy, and spheroid viability was assessed using a 3-(4,5-dimethylthiazol-2-yl)-2,5-diphenyltetrazolium bromide (MTT) assay. The spheroids treated with Gem-loaded MBs and then exposed to US displayed a loss of spheroid integrity compared to those treated with US on its own (i.e., without MBs) or with Gem-loaded MBs alone (i.e., without US). However, the acoustically mediated co-delivery of gemcitabine and paclitaxel into spheroids using Gem/PTX-loaded MBs magnified this biological response. Indeed, a significant loss of spheroid cohesion was detected with much cellular debris surrounding these spheroids. As a result, Gem-loaded MBs following US exposure reduced the spheroid viability from 90 ± 10% (i.e., Gem-loaded MBs without US) to 62 ± 5% (i.e., Gem-loaded MBs with US) while the combination of US with Gem/PTX-loaded MBs decreased this viability from 84 ± 10% to 30 ± 6%. Moreover, Logan et al. investigated the therapeutic efficacy of their therapeutic MBs in a subcutaneous pancreatic cancer (BxPC3) xenograft tumor model. They administered intravenously the Gem-loaded MBs (2.8 ± 0.3 mg/kg Gem; 1 × 10^9^ ± 2 × 10^7^ MBs/mL) or Gem/PTX-loaded MBs (3.2 ± 0.4 mg/kg Gem; 2.0 ± 0.2 mg/kg PTX; 8.6 × 10^8^ ± 1 × 10^7^ MBs/mL). Then, the tumors (150 mm^3^) were exposed to US (1 MHz, 30% DC, 100 Hz PRF, 0.48 MPa PNP) for 3.5 min, and their dimensions were measured using a Vernier caliper during the tumor growth. Regardless of the type of the therapeutic MBs used, this protocol inhibited the tumor growth compared to control conditions (no treatment, free Gem, or free Gem/PTX). Surprisingly, there was no significant difference in the therapeutic efficacy between both therapeutic MBs. As previously described for Gao et al., in vivo results partly confirmed the results obtained with the spheroids. In addition, the same limitations persist, namely the use of two different tumor cell lines in vitro and in vivo and the use of the avascular spheroids. Nevertheless, these data confirmed the potential of MB-assisted US to co-deliver both chemotherapeutic drugs for the targeted treatment of pancreatic cancer.
Evaluation of the impact of the tumor microenvironment on the efficacy of treatment

The importance of the tumor microenvironment in the process of tumor progression and in resistance to pharmacological treatments has been widely studied [2,4]. To mimic this tumor microenvironment and its heterogeneity, MCTSs are generated by coculturing tumor cells with other types of cells (e.g., fibroblasts, adipocytes, immune cells, etc.) and with extracellular matrix. For example, the coculture of fibroblasts with tumor cells induces their differentiation into cancer-associated fibroblasts, which are characterized by the expression of α-smooth muscle actin [74]. These cancer-associated fibroblasts promote tumor growth, tumor aggressiveness, metastasis, and resistance to anticancer drugs, partly due to the formation of a dense extracellular matrix that restricts drug penetration into the tumor, thereby rendering treatment less efficient [75,76]. In this context, MCTSs are also used to develop MB-assisted US as a targeted drug delivery method [48,70].

Roovers et al. [70] investigated and compared the efficacy of MB-assisted US using fluorescent regular liposomes (RL) and temperature-sensitive liposomes (TSL) based on Doxil^®^ and ThermoDox^®^ formulations to encapsulate and deliver Dox into spheroids and MCTSs. These formulations were either co-administered with MBs or loaded on MBs. To achieve this objective, murine breast cancer (4T1) cells only (spheroids) or with fibroblasts in a 1:5 ratio (MCTS) were seeded and cultured on an agarose-based microwell array for 48 h for spheroid formation (100–150 µm). The efficacy of Dox and fluorescent liposome (LPS) delivery was assessed 24 h after US exposure using flow cytometry and confocal microscopy (Figure 4). Flow cytometry analysis confirmed that MB-assisted US using RL-loaded MBs is a more efficient strategy to deliver Dox into spheroids and MCTSs. The gradient in fluorescence shown by the cell population thus treated indicated that the location of the cell within the spheroids and MCTSs significantly impacted the efficacy of Dox and/or RL uptake. In addition, the positive correlation between RL and Dox fluorescence revealed that intact RL-loaded LPS was delivered into the spheroids and MCTSs. As expected, confocal microscopy revealed a colocalization of RL and Dox fluorescence in the outer layers of the spheroids because of the slow Dox release from RL. Then, both 3D models were incubated with TSL-loaded MBs and then exposed to US. Twenty minutes and one centrifugation later, spheroids and MCTSs were heated at 42 °C for 15 min to induce Dox release from TSL. Flow cytometry analysis showed that the treatment of spheroids and MCTSs with TSL-loaded MBs and US, followed by mild hyperthermia is the more efficient method to deliver Dox from TSL. As a result, the correlation between TSL and Dox fluorescence was less notable. Confocal microscopy analysis confirmed these results. After mild hyperthermia, Dox leaked out of the TSL and diffused further into the spheroids as well as MCTSs, suggesting that the stroma in the MCTS is not a barrier to the diffusion of Dox into MCTSs.

Regardless of the type of Dox-loaded LPS, similar results were obtained between spheroids and MCTS, confirming that the tumor stroma has no significant impact on acoustically mediated Dox delivery.

Then, the cytotoxicity of these different strategies was evaluated 72 h after treatment by observing morphological changes in spheroids and MCTSs using optical microscopy and by assessing spheroid viability using a CellTiter-Glo^®^ 3D cell viability assay. Both untreated spheroids and MCTSs grew closely in size over time and were perfectly cohesive. The spheroids treated with TSL-loaded MBs and US, followed by mild hyperthermia, displayed a significant loss of cohesion, while MCTSs mostly shed cell fragments. This difference may be explained by the presence of extracellular matrix in the MCTS, which maintained MCTS cohesion. It is unfortunate that the authors did not report similar data for RL-loaded MBs. Nevertheless, they described the spheroid viability for both Dox-loaded MB formulations. Regardless of both 3D models, MB-assisted US using RL-loaded MBs induced significant cytotoxicity (viability < 30% for spheroids and MCTS) compared to RL treatment alone (viability > 60% for spheroids and MCTS) or to the coadministration of RL with MBs and exposure to US (viability > 50% for spheroids; >60% MCTS). This strategy is as effective as Dox treatment alone. Similar results were obtained when spheroids and MCTSs were exposed to TSL-loaded MBs and US, followed by mild hyperthermia.

Finally, the authors investigated the protective effect of the fibroblast-derived matrix on breast cancer cell viability by generating spheroids and MCTSs with luciferase-overexpressing breast cancer cells. Thus, luciferase expression was measured to selectively assess the viability of breast cancer cells in spheroids and MCTS 72 h after treatment. Regardless of LPS type, this analysis revealed that the MCTSs were less sensitive to Dox than spheroids when they were treated with Dox-loaded LPS (viability > 75% for MCTS versus <50% for spheroids) only or coadministered with Dox-loaded LPS and MBs and exposed to US (RL: viability > 75% for MCTS versus <50% for spheroids; TSL: viability > 50% for MCTS versus >40% for spheroids). This result thus confirms the protective effects of fibroblasts against Dox. However, MB-assisted US using both types of Dox-loaded MBs resulted in an identical decrease in cell viability into spheroids and MCTSs (viability < 25%), suggesting that this acoustically enhanced Dox delivery made it possible to overcome the protective effects of fibroblasts. This therapeutic efficacy was also higher than those of the treatments described above but as effective as free Dox (viability < 25%).

In conclusion, spheroids and MCTSs are powerful 3D tumor models to design and to validate protocols for drug delivery using MB-assisted US. Indeed, they are mainly used (i) to validate some proofs of concept, (ii) to investigate the effects of acoustic parameters and new formulations of MBs, (iii) to assess the penetration and accumulation of anticancer drugs (free or formulated drugs), (iv) to evaluate the therapeutic efficacy of these drugs, and (v) to study the influence of the tumor microenvironment on the efficacies of drug delivery and treatments. Nevertheless, their main limitation is the absence of tumor microvasculature, thus restricting the comparison of results on spheroids and MCTSs with in vivo data.

#### 3.2.2. Sonodynamic Therapy

Sonodynamic therapy (SDT) represents an emerging therapeutic strategy that offers the possibility of eradicating solid tumors in a noninvasive and targeted manner [77]. This strategy relies on the sensitization of target tumor tissues with a nontoxic sensitizing agent (e.g., 5-aminolevulinic acid, Rose Bengal, protoporphyrin IX, etc.) and subsequent exposure of the sensitized tissues to relatively low-intensity pulsed ultrasound (LIPUS) [78]. SDT uses nontoxic sensitizers, which are commonly exploited as photosensitizers in photodynamic therapy (PDT). In addition, LIPUS is a nontoxic and nonthermal stimulus. The strong tissue penetrating power of US (>1 cm) makes it a superior anticancer strategy compared to similar alternative strategies including PDT, in which less penetrating light sources (<1 cm) are employed to induce cytotoxic effects in sensitized tumor tissues. Thus, SDT would find wider oncological applications, specifically for the noninvasive and targeted treatment of deep tumor lesions. Both sensitization and US exposure are nontoxic alone, whereas their combination results in cytotoxic events through the generation of reactive oxygen species (ROS) [79]. This strategy offers the advantage of increasing on-target responses while reducing adverse effects compared to conventional chemotherapy. In addition, the efficacy of SDT is often evaluated in association with chemotherapy, PDT, or targeted therapies [32,80,81,82,83]. As previously described for acoustically mediated drug delivery, the efficacy and safety of SDT strongly depend on (i) the pharmacological properties of sensitizers [83,84,85], (ii) the physiology of tumor tissue [83], (iii) US devices and parameters [80], and (iv) SDT treatment schemes on their own or combined with other anticancer therapies, including chemotherapy, radiotherapy, PDT, and therapeutic ultrasound [80,81,86,87,88,89]. In the next section, we will present and discuss the studies that exploited spheroids and MCTSs for the design of SDT protocols (Table 3).
Investigation of drug penetration and efficacy

As previously reported, the in-homogeneous tumor vasculature and the increased interstitial pressure restricted the penetration and accumulation of NPs inside the tumor core [4,90,91]. Zhou et al. [86] investigated the efficacy of low-intensity focused ultrasound (LIFU) to enhance the intratumoral diffusion of NPs inside cervical cancer (HeLa) spheroids after one week of culture in spheroid microplates. In this study, the authors developed theranostic nanoparticles (NPs) loaded with a sonosensitizer, IR780, a magnetic resonance imaging contrast agent, gadolinium diethylenetriaminepentacetate, and a phase transition material, perfluorohexane, for the treatment of cervical cancer using SDT and acoustic droplet vaporization. Cervical cancer cells were seeded and cultured in ULA microplates. One week later, fluorescent NPs (2 mg/mL) were added to spheroids and exposed to LIFU (2 treatments with 10 min intervals: 2.5 W cm^−2^ for 20 s). Then, the spheroids were observed under CLSM at 6 h posttreatment. In the absence of LIFU exposure, the fluorescent NPs were preferentially located on the surface of spheroids, and only a few NPs were detected inside the spheroids. The acoustically mediated NP delivery revealed a significant increase in fluorescent NPs inside the spheroid core. These NPs were homogenously distributed inside the spheroids. To confirm these results in vivo, the NPs (5 mg/mL, 200 μL) were injected intravenously in a subcutaneous cervical cancer xenograft tumor model. Then, the tumors were exposed to LIFU (two treatments with 10 min intervals: 2.5 W cm^−2^ for 10 min) 6 h after the NP administration. Six hours later, magnetic resonance imaging confirmed that the acoustically mediated NP delivery was a promising strategy to enhance the concentration of NPs in the tumor core. As reported above, these in vivo data partly confirmed the in vitro data on the spheroids. In addition, we notice here again that the main limitation in this study is the exploitation of the avascular spheroids. Indeed, the spheroids were incubated directly with the NPs whereas NPs were administered intravenously in vivo. We can legitimately ask ourselves whether the LIFU treatment has an effect only on the intratumoral bioavailability. The LIFU may improve the enhanced permeability and retention (EPR) effect by permeabilizing the tumor microvasculature. In addition, as in other publications, the analysis of anticancer treatment efficacy on spheroids was restricted to a unique parameter, here, the increase in the quantity of NPs inside the tumor. Indeed, the therapeutic efficacy of these theranostic NPs was evaluated in a subcutaneous cervical cancer xenograft tumor model. Thus, the increase in therapeutic efficacy of NP after LIFU treatment was positively correlated with the increase in intratumoral bioavailability of NPs induced by LIFU.

Similarly, Sun et al. [85] investigated the penetration and the efficiency of liposomes (iRGD-Lipo-sinoporphyrin sodium) targeting integrin-overexpressing cancer cells through an internalizing-RGD peptide (iRGD) and containing a sonosensitizer, sinoporphyrin sodium, on spheroids of glioma (C6). These spheroids were generated using ULA microplates for one week. Free-form and liposomal formulations (bare and targeted) of sonosensitizer (1 μM) were incubated with spheroids for 4 h. Using the native fluorescence of sonosensitizer, the spheroids were then observed under CLSM. The free-form and the bare liposomes of the sonosensitizer were mainly located in the peripheral layers of spheroids, while the targeted liposomes of the sonosensitizer were uniformly distributed inside the spheroids. Subsequently, the therapeutic efficacy of these formulations (0.5 μM) for SDT (0.6 W cm^−2^ for 1 min) were assessed by observing the morphology of spheroids under optical microscopy 24 h after treatment. Untreated spheroids depicted uniform and cohesive morphology. Free form and bare and targeted liposomal formulations of sonosensitizer and US treatments alone did not affect the morphology of spheroids. SDT with the free form of sonosensitizer induced a decrease in the size of only a few spheroids. Using the bare liposomes of sonosensitizer, SDT caused a strong loss of spheroid cohesion, thus inducing abundant individual cells in the plate. Finally, the exposure of spheroids to SDT with targeted liposomes of sonosensitizer generated full destruction of all spheroids.

Altogether, these results on spheroids indicated that the targeted liposomes of the sonosensitizer significantly accumulated into spheroids, thus causing significant cell death. By choosing to exploit an orthotopic glioma xenograft tumor model and to a administer intravenously their targeted liposomes of the sonosensitizer, the authors were confronted with the blood–brain barrier, which restricts the intracerebral delivery of therapeutic molecules [92]. To overcome this limitation, which they have not encountered while exploiting spheroids, they delivered their targeted liposomes of the sonosensitizer using MB-assisted US (see Section 3.2.1). Lab-made phospholipid MBs (20 μL) were injected intravenously, and focused US (1 MHz, burst interval time: 1 s, 20% DC, 1 W, for 1 min) were immediately applied to tumors in order to open the blood–brain barrier. Then, the targeted liposomes of the sonosensitizer (1 mg/kg DVDMS) were administered intravenously. Thirty hours later, the SDT treatment (1 MHz, 1 W, for 1 min) was repeated twice within a five-day interval. This therapeutic protocol significantly prolonged the survival time (40 days versus 15 days) and improved the living status of glioma-bearing mice compared to control treatments (no treatment, free sonosensitizer, and SDT; untargeted liposomes of sonosensitizer and SDT). This improvement in therapeutic efficacy is closely associated with an increase in quantity of the sonosensitizer using MB-assisted US in the tumor tissue.

**Table 3 pharmaceutics-15-00806-t003:** Sonodynamic therapy.

Ref.	Sonosensitizer	Cells	Spheroid Formation Method	US Parameters	Specific Assay	Cytotoxicity Assay	Main Outcomes
[86]	Theranostic nanoparticles (Dil-IGP@P NPs): IR780 as sonosensitizer and photoacoustic agent Perfluorohexane for acoustic droplet vaporization	HeLa cells (cervical cancer cell line)	Spheroid microplate	2.5 W cm^−2^ for 20 s, with two treatments at 10 min interval	Penetration of fluorescent nanoparticles using confocal microscopy	NA	Strong and uniform distribution of nanoparticles inside the spheroids
[83]	Therapeutic nanoplatform (CEPH): Hematoporphyrin as sonosensitizer EGFR-TK inhibitor as targeted molecular therapies Perfluorooctyl bromide for oxygen delivery	A549 - PC-9 (NSCLC cell lines)	Hydrogel of gelatin methacryloyl	0.1 W cm^−2^ for 1 min	Assessment of oxygen condition using fluorescent hypoxia probe	NA	Baseline spheroid hypoxia increased with SDT due to the oxygen-consuming nature of the process, but hypoxia is reduced after CEPH and US treatment
	Therapeutic nanoplatform (RGD/PTK@PEG/Dox): Porphyrin as sonosensitizer Doxorubicin (Dox) as anticancer drug RGD peptide as targeting agent	MCF-7 (BC cell line)	Methylcellulose	1 MHz, 2.0 W cm^−2^ for 15 min	Dox delivery using confocal microscopy	Live/dead (calcein-AM/propidium iodide (PI)) cell assay	Increase in Dox uptake inside the spheroid and specifically in deeper region Increase in cell mortality inside the spheroid
[32]	Chlorin-e6 as sonosensitizer loaded on graphen nanoribbons	SKOV-3 (Ovarian cancer cell line)	polyHEMA-coated plate - Coculture of spheroid with the monolayer LP-9 mesothelial cells	1 MHz, 0.8 Wcm^−2^ for 30 s	Detection of ROS generation with DCFDA assay	Live/dead cell assay	Strong production of ROS in the spheroids Significant increase in cell death in the spheroids Neither ROS production nor cytotoxic effect for LP-9 cell
[85]	Liposomes (Lipo): Sinoporphyrin sodium (DVDMS) as sonosensitizer iRGD as targeting agent	C6 (Murine glioma cell line)	ULA plate	0.6 W cm^−2^	NA	Spheroid damage using optical microscopy	Spheroid size reduction with DVDMS treatment alone + US compared to all other conditions without US (i.e., no treatment, DVDMS alone, Lipo-DVDMS, and iRGD-Lipo-DVDMS) Complete spheroid destruction with Lipo-DVDMS + US or with iRGD-Lipo-DVDMS + US
[80]	AlPcS_2a_ as sonosensitizer Bleomycin (BLM) as anticancer drug	F98 (Glioma cell line)	ULA plate	1 MHz, 100% DC, 0–0.6 W cm^−2^ for 3 min	Disruption of endolysosome membrane labelled with Lysotracker using fluorescence microscopy	Measure of spheroid volume Live/dead (Hoechst/ethidium homodimer 1) cell assay	After US treatment, Lysotracker escape from endosomes Sonochemical internalization (SCI) have higher growth-inhibiting effect on spheroid than SDT or BLM alone Increase in the fluorescence of ethidium homodimer (dead cells) after SCI treatment compared to BLM alone or SDT alone
[87]	5-aminolevulinc acid as sonosensitizer	A2058 (Melanoma cell line) - HT-1080 (Fibrosarcoma cell line) - SH-SY5Y (Neuroblastoma cell line)	Hanging drop plate (Perfecta 3D)	A2058: 0.32 mJ mm^−2^ for 1000 impulses at 4 impulses s^−1^ - HT-1080 and SH-SY5Y: 0.43 mJ mm^−2^ for 500 impulses at 4 impulses s^−1^	NA	Monitoring of spheroid volume	Significant reduction in the spheroid volume from 24 h for HT-1080 and SH-SY5Y spheroids, and from 48 h for A2058 spheroids
[81]	Lab-made lipid-shelled MBs: (1) O_2_MB-PTX-RB Rose Bengal as sonosensitizer Paclitaxel (PTX) as anticancer drug (2) O_2_MB-PTX-Dox Dox and PTX as anticancer drugs	MCF-7 (BC cell line)	ULA plate	1 MHz, 100 Hz PRF, 50% DC, 3.0 W cm^−2^ for 30 s	NA	MTT assay Spheroid damage PI staining of dead cells	Significant decrease in cell viability Significant reduction in spheroid size
[82]	Nanoplatforms: Cu-CuFe_2_O_4_ as sonosensitizer Dox as anticancer drug	MCF-7	NA	2.0 W cm^−2^ for 15 min	NA	Live/dead (calcein-AM/PI) cell assay	Dox-loaded Cu-CuFe_2_O_4_ NPs + US induced a strong PI fluorescent signal on the whole spheroid compared to US only and Cu-CuFe_2_O_4_ NPs without US
[88]	Poly(lactic-*co*-glycolic acid) nanospheres (Chlor-PLGA NCs) Chlorophyll (Chlor) as sonosensitizer	DU-145 (Human PDAC cell line)	Plate coated with agarose	1.5 MHz, 1.5 W cm^−2^ for 5 min	NA	PI staining of dead cells Measure of spheroid volume	Significant increase in cell mortality Significant reduction in spheroid volume
[89]	Poly-methyl methacrylate core-shell nanoparticles: Meso-tetrakis (4-sulfonatophenyl) porphyrin as sonosensitizer	SH-SY5Y	NA	0.43 mJ mm^−2^ for 500 impulses (4 impulses s^−1^)	NA	Monitoring of spheroid volume	Significant reduction in spheroid volume

Dil-IGP@P NPs = 1:1′-Dioctadecyl-3,3,3′,3′-tetramethyl indocarbocyanine perchlorate—I(IR780)G(gadolinium diethylenetriaminepentacetate)P(perfluorohexane)@P(lactic-co-glycolic acid) nanoparticles; NA = not available; CEPH = erlotinib modified chitosan perfluorooctyl bromide hematoporphyrin; EGFR-TK = epidermal growth factor receptor tyrosine kinase; NSCLC = nonsmall cell lung cancer; RGD/PTK@PEG/Dox = Arg-Gly-Asp peptide (RGD)/porphyrin, platinum, thioketal linker(PTK@)/polyethylene glycol (PEG)/doxorubicin (Dox); Dox = doxorubicin; BC = breast cancer; PI = propidium iodide; polyHEMA = poly(2-hydroxyethyl methacrylate); ROS = reactive oxygen species; DCFDA = 2′,7′-dichlorofluorescin diacetate; DVDMS = sinoporphyrin sodium; iRGD = internalizing RGD; AlPcS2a = aluminum phthalocyanine disulfonate; ULA = ultralow attachment; DC = duty cycle; SC I = sonochemical internalization; BLM = bleomycin; MBs = microbubbles; PTX = paclitaxel; RB = Rose Bengal; PRF = pulse repetition frequency; MTT assay = 3-(4,5-dimethylthiazol-2-yl)-2,5-diphenyltetrazolium bromide assay; Chlor-PLGA NCs = chlorophyll-containing poly(lactic-co-glycolic acid) nanospheres and nanocapsules; PDAC = pancreatic ductal adenocarcinoma.

Afterward, Zhao et al. [84] designed ROS-responsive nanoscale coordination polymers through the self-assembly of porphyrins (sonosensitizer) and platinum, which carried an RGD targeting peptide and ROS-cleavable linker to increase Dox release during SDT (1.0 MHz, 2.0 W cm^−2^ for 15 min). The penetration and therapeutic efficacy of this therapeutic nanoplatform were evaluated on spheroids of breast cancer (MCF-7). These spheroids were generated in medium containing methylcellulose for 2 days. To investigate the Dox release and uptake behavior inside the spheroids, they were incubated with the targeted nanoplatform (200 μg/mL), exposed to US, and finally observed under CLSM. The spheroids incubated with the targeted nanoplatforms only showed a weak fluorescence intensity of Dox on their outer surface, while Dox was efficiently released and deeply internalized in the core of spheroids after US exposure. To assess the therapeutic efficacy of SDT, the spheroids were incubated with the targeted nanoplatform. Four hours later, they were stained using a live/dead assay (calcein-AM/PI). Subsequently, the spheroids were exposed to US. SDT significantly increased the apoptosis of tumor cells compared to treatment with the targeted nanoplatform only. In conclusion, this controllable and stimuli-responsive therapeutic nanoplatform is promising for SDT.

Moreover, Lee et al. [32] developed an SDT strategy on metastatic ovarian cancer spheroids (SKOV-3) using graphene nanoribbons (GNRs) functionalized with 4-arm polyethylene glycol (PEG) and chlorin e6 (Ce6) as a sonosensitizer (GNR-PEG-Ce6). To generate these spheroids, ovarian cancer cells were seeded and cultured on a nonadhesive pHEMA surface for 24 h (30–200 µm spheroid size). Optical imaging revealed that the adsorption of GNR-PEG-Ce6 (50 and 100 μg/mL) at the surface of spheroids prevented their adhesion to extracellular matrix proteins. This loss of spheroid adhesion was associated with a significant downregulation of integrin β1 and CD44 proteins on the surface of spheroids. This incubation also induced a loss of spheroid cohesion related to a significant downregulated expression of E-cadherin. Moreover, GNR-PEG-Ce6 (50 μg/mL) fully inhibited the adhesion and spreading of spheroids onto the mesothelial layer (LP-9) and subsequent mesothelial clearance, a key metastatic process. GNR-PEG-Ce6 can kill ovarian cancer spheroids adhered to mesothelial cell monolayers when combined with US. The interaction with GNR-PEG-Ce6 also loosens intercellular adhesions within the spheroids, rendering them more susceptible to chemotherapeutic drugs including cisplatin and paclitaxel. Thus, GNR-PEG-Ce6 increases the efficacy of chemotherapeutic and sonodynamic combination therapies. To assess the therapeutic efficacy of SDT using GNR-PEG-Ce6, spheroids were placed on a mesothelial cell monolayer and treated with GNR-PEG-Ce6 (50 μg/mL) for 48 h (Figure 5A). They were exposed to US (1 MHz, 0.8 W cm^−2^) for 30 s. Then, a live/dead assay (calcein-AM/ethidium bromide) was performed 30 min later under fluorescence microscopy. The qualitative analysis of microscopic images showed that SDT using GNR-PEG-Ce6 induced a significant loss of spheroid viability compared to SDT using GNR-PEG (Figure 5B). Using a 2′,7′-dichlorofluorescin diacetate cellular ROS assay, the authors demonstrated that this cytotoxicity was related to a 3.5-fold increase in ROS levels after SDT with GNR-PEG-Ce6 compared to SDT with GNR-PEG for SKOV-3 spheroids but not in mesothelial cells (Figure 5C,D). In addition, using the live/dead assay, they reported that chemotherapy alone with cisplatin (20 μM) or paclitaxel (20 μM) led to a significant decrease in the viability of GNR-PEG-Ce6-treated spheroids, while the same chemotherapy dose on its own did not induce a notable effect on spheroid viability. These results suggest that the loss of intercellular interaction in GNR-PEG-Ce6-treated spheroids improves the penetration of chemotherapeutic drugs inside the spheroids and thus their therapeutic efficacy. Finally, Lee et al. also validated the therapeutic benefit of this SDT on tumor spheroids derived from the ascites fluid of ovarian cancer patients.
Influence of the tumor microenvironment

As previously reported, tumor hypoxia can limit the efficacy of SDT, which is highly oxygen consuming. To overcome this issue, Zhang et al. [83] designed a multifunctional chitosan-based nanoplatform (CEPH), which specifically targets EGFR-overexpressing cancer cells with erlotinib, an epidermal growth factor receptor tyrosine kinase inhibitor (EGFR-TK inhibitor) and is loaded with a sonosensitizer, hematoporphyrin and perfluorooctyl bromide, an oxygen-storing agent, to locally deliver hematoporphyrin and oxygen to EGFR-overexpressing NSCLC cells. The effects of SDT (0.1 W cm^−2^ for 1 min) using CEPH (50 μg/mL) on tumor hypoxia were investigated using NSCLC spheroids. NSCLC cells (PC-9) were seeded and cultured using a protocol based on photocurable hydrogels for 2 weeks. Then, they were incubated under hypoxic conditions for 24 h. The spheroids were treated with CEPH for 2 h and exposed to US for 1 min. Subsequently, a fluorescent hypoxia probe (Image-iTTM green hypoxia agent) was exploited to assess the oxygen condition inside the spheroids. The untreated spheroids depicted an intense hypoxia signal, while their incubation with CEPH only significantly decreased this hypoxia signal. SDT consumes oxygen, thus increasing hypoxia. Nevertheless, the treatment of spheroids with CEPH in combination with US significantly decreased the hypoxia inside the spheroids compared to untreated spheroids. These results confirm that SDT is an oxygen-consuming process. This new therapeutic nanoplatform could alleviate the effects of tumor hypoxia and SDT-induced hypoxia as well as increase SDT efficacy. Unfortunately, the authors only exploited spheroids to investigate the effects of SDT on tumor hypoxia over a short post-treatment period. Indeed, they demonstrated that the combination of CEPH with US inhibited NSCLC cell growth under normoxic and hypoxic conditions and increased the synergistic effects of SDT and targeted molecular therapy using a monolayer of NSCLC cells.

In conclusion, in vitro studies have successfully reported the exploitation of spheroids to design SDT strategies in oncology. Thus, new sonosensitizers have been validated on spheroids and MCTSs, but the influences of the tumor microenvironment on the therapeutic efficacy of SDT have also been investigated in these 3D models. In addition, the therapeutic benefit of SDT combined with chemotherapy has also been described in spheroids and MCTSs. Even if these SDT strategies were also validated in small animal models of primary tumors and metastases of tumors, the absence of tumor vasculature in the spheroids and MCTSs remains their main limitation. The EPR effect of sonosensitizer-loaded nanoparticle cannot be investigated on avascular spheroids and MCTSs.

#### 3.2.3. Other Types of US-Mediated Drug Therapies

More anecdotally, spheroids have also been exploited to develop other acoustically mediated drug delivery strategies using US-induced inertial cavitation, low-intensity US (LIUS), laser-generated focused US (LGFU), and US-induced mild hyperthermia (Table 4) [93,94,95,96].
US-induced inertial cavitation

The tumor microenvironment of pancreatic ductal adenocarcinoma contributes to chemoresistance. Leenhardt et al. [30] hypothesized that US-induced inertial cavitation could induce the disruption of the tumor microenvironment, thus facilitating the intratumoral bioavailability of chemotherapeutic drugs and improving their therapeutic efficacy. To validate this hypothesis, they exploited MCTSs generated from murine pancreatic ductal adenocarcinoma cells and murine embryonic fibroblasts using the magnetic nanoshuttle method. As we previously described, fibroblasts produce type 1 collagen, a major component of the extracellular matrix, which makes up the tumor microenvironment. The MCTSs were exposed to US with incremental inertial cavitation indices (1.1 MHz, 100 Hz PRF, 25% DC, 0.4 to 3.5 MPa for 20 s) in the presence of gemcitabine (5 μM). Spheroid viability was evaluated using resazurin and BV-510 assays. The MCTSs were significantly less sensitive to gemcitabine (5 μM) than spheroids without fibroblasts (spheroid viability of 80% versus 60%), thus confirming that the tumor microenvironment restricted the therapeutic efficacy of gemcitabine. However, the combination of US-induced inertial cavitation (cavitation index of 20) with gemcitabine induced a significant decrease in MCTS viability compared to gemcitabine treatment alone (MCTS viability of 74.7 ± 5.5% versus 90.8 ± 5.5%). Both inertial cavitation (cavitation index of 20) and gemcitabine (5 μM) treatments altered the viability of tumor cells (cell viability < 20% versus 30%) compared to untreated MCTSs (cell viability of 50%). However, these treatments had no effect on fibroblast viability, thus supporting the protective role of fibroblasts against tumor cells. Altogether, these results suggest that the combination of US-induced cavitation with gemcitabine enhanced the therapeutic efficacy of gemcitabine in this MCTS model, thus validating the hypothesis that inertial cavitation may overcome stromal-related chemoresistance. Further experiments investigating the influence of inertial cavitation on the extracellular matrix of MCTSs and on gemcitabine penetration inside the MCTS are required to fully validate this hypothesis.
pharmaceutics-15-00806-t004_Table 4Table 4LIPUS and gold NPs, LGFU, acoustic cavitation, and mild hyperthermia.Ref.Drug/Dye/NPs/LiposomeCell LineSpheroid Formation MethodUS ParameterAnalysisMain Outcomes[97]Retinoid acid (RA)Temozolomide-loaded gold nanoparticles (TGNPs) as anticancer drugsU87-MG (GBM cancer cell line)Forced floating methods666 and 1066 kHz dual- frequency, 20% DC, 80 kPa for 10 min/day for 3 daysMTT assayLive/dead cell assayLDH assayApoptosis (Annexin V/PI) assayMorphologyCombined treatment of RA and LIPUS-mediated TGNP induced:Significant increase in the number of dead cells through apoptosis processSignificant decrease in the size and number of spheroidsMajor structural damages[93]Cisplatin (Cis) as anticancer drugGold nanocones (AuNCs)A2780 (sensitive to cisplatin)-A2780cis (resistant to cisplatin)(ovarian cancer cell lines)Agarose coated plate with Polyethylene glycol and dextraninterface1 MHz, 50% DC, 1.0 W cm^−2^ for 3 minCalcein cell stainingSignificant decrease in the viability of A2780 and A2780cis spheroidsAuNCs improve the therapeutic efficacy of US+Cis treatment[96]Doxorubicin (Dox) loaded on Poly(lactic-co-glycolic acid) (PLGA) embedded in alginate microgelsHeLa cells (cervical cancer cell line)Liquid overlay method withAgarose-gel-coated plate18 mJ for 5 minMonitoring of spheroid volumeSignificant spheroid growth inhibition[30]Gemcitabine (Gem)KPC (Murine PDAC cancer cells)-iMEF (Murine embryonic fibroblasts)Magnetic 3D bioprinting protocol using nanoshuttlesCell coculture1.1 MHz frequency, 100 Hz PRF, 25% DC, 0.4 to 3.5 MPa PNP for 20 sResazurin assay BV-510 viability assaySignificant reduction in spheroid sizeSignificant decrease in the viability of KPC cells but not that of iMEF cellsMCTS are less sensitive to low Gem dose compared to spheroids[33]Echogenic low-temperature-sensitive liposomes (E-LTSLs) Dox as anticancer drugA549 (NSCLC cells)Liquid overlay method withagarose-gel-coated plate5 MHz PRF, 50% DC, 3.25 W total acoustic power for 60 sConcentration of Dox released in the supernatant and inside the spheroid after lysis, evaluate by spectrophotometryIncrease in Dox concentration in cell medium to 34 µM for E-LTSL + HIFU compared to 12 µM without HIFU Cellular uptake of Dox was 9.8 µM for E-LTSL + HIFU compared to 2.9 µM without HIFU[98]Traditional temperature-sensitive liposomes (TTSLs) Dox as anticancer drugU87-MGHanging drop method1 MHz sinusoidal waves with a PRP of 1 ms, 400 cycles per pulse and for different acoustic pressures (i.e., 0–2 MPa) and total exposure time (i.e., 0–30 min)Spheroid size, doubling time
US parameters at 1.75 MPa during 10 min were optimal to induce Dox release from TTSL2-fold doubling time increase was observed for spheroids treated with TTLS and FUS compared to spheroids treated with TTLS without hyperthermia inductionRA = retinoic acid; TGNPs = temozolomide-loaded gold nanoparticles; GBM = glioblastoma multiforme; DC = duty cycle; MTT assay = 3 -(4,5-dimethylthiazol-2-yl)-2,5-diphenyltetrazolium bromide assay; LDH = lactate dehydrogenase; PI = propidium iodide; LIPUS = low-intensity pulsed ultrasound; Cis = cisplatin; AuNCs = gold nanocones; Dox = doxorubicin; PLGA = poly(lactic-co-glycolic acid); Gem = gemcitabine; PDAC = pancreatic ductal adenocarcinoma; iMEF = murine embryonic fibroblast; KPC = murine PDAC cancer cells; PRF = pulse repetition frequency; PNP = peak negative pressure; E-LTSL = echogenic low-temperature-sensitive liposome; NSCLC = nonsmall cell lung cancer; HIFU = high intensity focused ultrasound; TTSL = traditional temperature-sensitive liposome; PRP = pulse repetition period.
Low-intensity ultrasound (LIUS)

As previously reported, cancer stem cells are undifferentiated and self-renewing cells that contribute to tumor initiation, progression, recurrence, and metastasis and resistance to anticancer treatments. For a few decades, new therapeutic strategies, also termed induction therapy, have aimed to suppress the protumor properties of cancer stem cells (self-renewal ability, the expression of stem-cell- and drug-resistance-related genes, etc.) and increasing anticancer drug sensitivity. Thus, Fadera et al. [97] designed an induction therapy based on retinoic acid (RA) with temozolomide (TMZ)-loaded gold NPs (TGNPs) associated with LIUS on glioblastoma multiforme (GBM) stem cells. TMZ is a current anticancer drug used for the treatment of GBM. RA is a typical chemical agent used to differentiate tumor stem cells by inhibiting signaling pathways involved in the induction and maintenance of the specific properties of cancer stem cells. According to the US parameters, LIUS also induces the differentiation of tumor stem cells. In this study, spheroids of GBM cells (U87-MG) were generated using a polyelectrolyte multilayer nanofilm system. This system selected GBM stem-like cells and enriched the GBM spheroids with them. The combinational therapy consisted of RA (unknown concentration) with TGNPs (1:15 ratio of TMZ and GNPs) combined with LIUS (1066 and 666 kHz dual frequency, 20% DC, 80 kPa for 10 min a day for a total of 3 days). This therapeutic efficacy was evaluated using different spheroid viability assays, including live/dead (calcein-AM/ethidium bromide), apoptosis (FITC-annexin V/PI), MTT, and lactate dehydrogenase assays. Regardless of the spheroid viability assays, the combinational therapy significantly decreased spheroid viability compared to TGNPs, free TMZ, or free RA treatments. For example, live/dead assay revealed that combinational therapy induced a two-fold decrease in spheroid viability compared to TGNP treatment alone. This therapy also resulted in a decrease in the size and number of spheroids. In addition, the exploration in CD133 expression, a biomarker of GBM stem-like cells, using flow cytometry, demonstrated that the combinational therapy led to a three-fold reduction of CD133 expression compared to untreated spheroids. Even though the molecular mechanisms governing the effectiveness of the combinational therapy have not been elucidated, these data suggested that this therapy can significantly decrease the viability of GBM cells as well as GBM stem-like cells.
Laser-generated focused ultrasound (LGFU)

Laser-generated focused ultrasound (LGFU) has been designed to perform high-frequency (>10 MHz) and high-precision (focal spot < 1 mm) US therapy with reduced US-induced heating (DC < 0.001%). Di et al. [96] hypothesized that this US technology is a promising strategy to stimulate the release of drugs from NPs. To validate this hypothesis, they assessed the capability of LGFU to trigger Dox release from alginate microgels encapsulated with Dox-loaded polymeric NPs and its anticancer efficacy toward cervical cancer (HeLa) spheroids (Figure 6). These spheroids were generated using liquid overlay methods and treated seven days later. A microgel solution was exposed to LGFU (18 mJ laser input for 30 s), and the supernatant (50 μL) was collected and applied to spheroids. LGFU induced a two-fold increase in the release of Dox from microgels compared to untreated microgels (i.e., passively released Dox). The therapeutic efficacy of released Dox was assessed by measuring the size of spheroids over time under optical microscopy. Untreated spheroids were increasingly cohesive over time and grew exponentially (spheroid volume of 70 × 10^4^ μm^3^ at Day 7), while spheroids treated with passively released Dox exhibited a few dead cells around them, and their volumes were still stable (spheroid volume of 15 × 10^4^ μm^3^ at Day 7), demonstrating a cytostatic effect of the passively released Dox. However, the exposure of spheroids to LGFU-released Dox induced a significant decrease in spheroid volume (spheroid volume < 5 × 10^4^ μm^3^ at Day 7) with a full loss of spheroid cohesion and the spheroids compared to both experimental conditions described above. In conclusion, LGFU successfully induced the release of Dox from Dox-formulated microgels, which significantly hampered spheroid growth.
Ultrasound-mediated mild hyperthermia

Focused ultrasound (FUS) is also a promising technology for the noninvasive and local mild heating (40–43 °C) of target tissues by depositing high acoustic intensity in the focal volume [99,100]. This mild hyperthermia can enhance the extravasation and accumulation of drug-loaded NPs within the heated tissue through increased blood perfusion and vascular permeability [19,101]. In addition, FUS-mediated mild hyperthermia also serves as an external trigger for targeted drug release from temperature-sensitive nanoparticles [102,103]. As previously described for other US therapies, spheroids were exploited here to evaluate new formulations of temperature-sensitive liposomes.

Indeed, Maples et al. [33] designed an echogenic and low-temperature-sensitive liposome (E-LTSL) loaded with Dox and perfluoropentane (US contrast agent), i.e., an US- imageable formulation of Dox-loaded TSL. They used spheroids of human lung adenocarcinoma (A549) to investigate the intracellular uptake of Dox released from E-LTSL after the application of FUS-mediated mild hyperthermia (41 °C). These spheroids were generated using a liquid overlay technique. Two or three days later, the spheroids were incubated with E-LTSL (100 μM) and then exposed to FUS (1.5 MHz, 5 MHz PRF, 5.5 W electric power, 3.25 W total acoustic power, 50% DC for 60 s). After FUS treatment, the supernatants were collected, and the spheroids were washed and then lysed to measure Dox fluorescence using a spectrofluorometer. The combination of E-LTSL and FUS-mediated mild hyperthermia resulted in complete release of encapsulated Dox in the supernatant compared to E-LTSL treatment alone (34.15 ± 0.2 μM versus 12.52 ± 0.08 μM). In agreement with these results, this treatment significantly increased the Dox concentration inside the spheroids in comparison with the LTSL treatment alone (9.85 ± 0.1 μM versus 2.97 ± 0.01 μM). Unfortunately, the authors did not monitor the cytotoxic effects of Dox on the growth of spheroids but instead on monolayers of human lung adenocarcinoma cells. They thus demonstrated the therapeutic efficacy of the combination of E-LTSL and FUS-mediated mild hyperthermia.

Similarly, Escoffre et al. [98] exploited glioblastoma spheroids (U-87 MG) to investigate the cytotoxic effect of Dox released from traditional temperature-sensitive liposomes (TTSLs) after FUS-mediated mild hyperthermia. An in vitro study using spectrofluorimetry showed that 85% of Dox was released from TTSLs (10 μM) when they were heated to 42 °C with FUS (1 MHz, 1 ms PRP, 400 cycles/pulse, 1.75 MPa PNP for 10 min), while no Dox was released from TTSLs at 37 °C. After FUS exposure, the solution of Dox was deposited on a glioblastoma cell monolayer to assess the intracellular uptake and cytotoxic effect of Dox using flow cytometry and MTT assays, respectively. This FUS-released Dox was as efficiently taken up by tumor cells as free Dox at 37 °C and induced a 50% decrease in cell viability compared to treatment with preheated TTSL at 37 °C. Then, the therapeutic efficacy of FUS-released Dox was evaluated on glioblastoma spheroids by measuring their growth under optical microscopy. The spheroids were generated using the hanging drop method. The FUS-released Dox resulted in a 1.7-fold decrease in spheroid growth in comparison to treatment with preheated TTSL at 37 °C. If this result confirmed the data obtained on glioblastoma cell monolayer, it is regrettable that the Dox penetration and accumulation inside the spheroids have not been investigated.

In conclusion, these in vitro studies have clearly illustrated that avascular spheroids and MCTSs are relevant biological tools for designing and validating other acoustically mediated drug therapies. They also make it possible to address the same questions (i.e., drug penetration, therapeutic efficacy, tumor microenvironment, etc.) encountered in drug delivery using MB-assisted US and SDT.

## 4. Future Perspectives and Conclusions

Tumor spheroids and MCTSs are relevant and promising 3D in vitro tumor models for assessing drug screening, drug design, drug targeting, drug toxicity, and drug delivery methods. Currently, several spheroid formation methods have been developed to readily and reproducibly generate spheroids and MCTSs. The most frequently employed method in the studies reviewed here is the liquid overlay method with an agarose coating. Additionally, we should note that some cell types are challenging to grow in 3D using simple methods and require either coculture with another cell type and/or with an extracellular matrix [104,105]. Some of these methods can be very expensive and require specific equipment but provide more sophisticated and specialized 3D tumor models, which are intermediate and complementary tools to cell culture monolayers or suspensions (2D in vitro tumor models) and to animal models.

For a few years, spheroids and MCTSs have been increasingly exploited to design and validate acoustically induced drug therapies, including drug delivery using MB-assisted US, US-induced inertial cavitation, LIUS, LGFU, US-induced mild hyperthermia, and SDT. Indeed, they are mainly used (i) to assess the effects of acoustic parameters on drug delivery and efficacy, (ii) to evaluate new drug formulations (e.g., drug-loaded NPs or MBs), (iii) to measure the penetration and distribution of anticancer drugs or sonosensitizers (free or formulated ones), (iv) to evaluate the therapeutic efficacy of these therapeutic molecules, and (v) to study the influence of the tumor microenvironment on the delivery of anticancer drugs or sonosensitizers and their therapeutic efficacies. One of the arguments for using spheroids rather than cell suspensions or cell culture monolayers is that they make it possible to investigate drug bioavailability and efficacy in 3D models similar to in vivo tumors. However, this literature review reveals that 33% of these studies designing acoustically mediated drug therapies mainly used spheroids to assess the penetration and biodistribution of free or formulated anticancer drugs/sonosensitizers. Indeed, the therapeutic efficacy of these therapeutic molecules on spheroid growth/viability was not investigated, whereas the dimensions of spheroids can easily be monitored over time under optical microscopy, or spheroid viability can be assessed using a dedicated 3D cell viability assay. Conversely, 67% of these studies did not explore the accumulation and bioavailability of free or formulated anticancer drugs/sonosensitizers inside the spheroids but only the therapeutic efficacy of these therapeutic molecules. This finding can easily be explained by the fact that the labeling of drugs or their nanoparticles with a fluorescent agent, for example, could affect their native penetration and biodistribution. Nevertheless, new analytical tools (e.g., immunoassays, mass spectrometry, etc.) will overcome this issue by dosing native drugs or nanoparticles.

The available literature claimed that spheroids and MCTSs can mimic the 3D architecture of tumors but also tumor heterogeneity (e.g., tumor cells, fibroblasts, adipocytes, immune cells, etc.) and microenvironment (e.g., intercellular interactions, extracellular matrix, hypoxia, necrosis, etc.), which influence the intratumoral bioavailability, pharmacokinetics, and pharmacodynamics of anticancer drugs and therefore chemoresistance [106,107]. Nevertheless, as with any model, they have some limitations and cannot fully reflect the reality of an in vivo tumor [108,109]. Indeed, spheroids and MCTSs can model the tumor heterogeneity and microenvironment to a limited extent only. In vivo, the cellular and molecular compositions of the tumor microenvironment are much more complex than those of in vitro 3D tumor models [108,109]. Supporting this aspect, this literature review highlighted the importance of using MCTSs, i.e., spheroids made up of tumor cells and other cell types (e.g., fibroblasts, cancer stem cells) in the development of acoustically mediated drug therapies. These MCTSs recreate a tumor environment responsible for resistance to drug therapies, which can be overcome by the active delivery of drugs and specifically acoustically mediated ones. However, the exploitation of MCTSs generated from patient-derived cancer cells (PDCs) could be a solution (but certainly imperfect one) in order to closely approximate a tumor microenvironment in vivo. The current data also showed that such MCTSs are not usual. Indeed, only one publication described the validation of acoustically mediated drug therapy (i.e., SDT) on spheroids derived from PDC (i.e., ascites fluid of ovarian cancer patients), while these PDC have been shown to reflect patient tumor characteristics (i.e., genetic and histological features of the original tumor) and clinical responses to anticancer treatments [110]. For these reasons, future studies on the performance of treatment including acoustically mediated drug therapies should use these PDC-derived spheroids.

In addition, some examples, where the authors confirmed their data obtained on spheroids or MCTSs, in subcutaneous or orthotopic tumor xenograft models, revealed that the lack of vessels in spheroids and MCTSs is still a major drawback for the full design and validation of acoustically mediated drug therapies [31,71,81,85,86]. Indeed, the spheroids were incubated directly with the free drugs or drug-loaded sonosensitive particles (e.g., MBs and liposomes), whereas these latter were administered intravenously in vivo. In the spheroid model, US treatment alone (e.g., inertial cavitation, LIFU, mild hyperthermia) or MB-assisted US may induce (i) the release of the drugs from sonosensitive particles; (ii) the loss of spheroid cohesion, thus facilitating the drug penetration inside the spheroids; and (iii) the permeabilization the tumor cells, thus improving the intracellular uptake of drugs. In the in vivo tumor model, US treatment can permeabilize the tumor microvasculature and consequently enhance both the drug extravasation and its intratumoral bioavailability. This permeabilization may increase the intracellular delivery of drug in the endothelial cells, thus potentiating the destruction of tumor vasculature and reducing the nutrient supply of tumors. In addition, the thermal and mechanical stimuli generated using the US treatment can also induce the release the drugs from sonosensitive particles in the tumor interstitium and permeabilize the tumor cells and as described regarding spheroids and MCTSs. However, MB-assisted US has no direct effect on tumor cells in vivo because the MBs are pure vascular agents. Nowadays, the intravenous route is the relatively easy and safe way to be used in the clinic for the administration of drugs. To partially overcome this limitation, the exploitation of 3D vascularized tumor spheroid-on-chip models could be an option. These models designate a microfluidic spheroid model, which moderately reflects the 3D architecture of the tumor, tumor heterogeneity, and microenvironment [111,112,113]. Unlike spheroids on their own, they are able to mimic the dynamic properties of the tumor microenvironment, for example, by ensuring the constant perfusion of cell medium (or any other relevant biological fluids) to supply nutrients, US contrast agents (e.g., MBs, nanobubbles, nanodroplets, etc.), and/or drugs (e.g., free and formulated anticancer drugs/sonosensitizers) and for removing waste. These models make it easy and precise to control the mechanical pressures (i.e., flow rate, shear stress, etc.) and the biochemical environment (e.g., pH, growth factors, hormones, etc.). Surprisingly, no 3D vascularized tumor spheroid-on-chip model has been used in the developments of acoustically mediated drug therapies. One of the most likely reasons could be that the biocompatible materials used for these spheroid-on-chips are not acousto-compatible. Therefore, it will be necessary to acoustically characterize these chip platforms to limit the formation of standing waves during their use for designing acoustically mediated drug therapies. The presence of significant standing waves will require adjustments of US setups or even modifications of materials on spheroid on chips. We will note that exploring the off-targeted effects of systemic treatments, including the acoustically mediated drug therapies, cannot be assessed with such a 3D vascularized tumor spheroid-on-chip model. Indeed, these effects are related to the interaction of the drugs with healthy tissues, which is itself influenced by the intratumoral bioavailability, pharmacokinetics, and pharmacodynamics of drugs.

Finally, another argument encouraging the exploitation of spheroids is the reduction in the number of animals or even the replacement of in vivo models. Unfortunately, we have no idea today whether the use of spheroids and MCTSs has made it possible to achieve these objectives of the 3R rules in the ethics of animal experimentation. Even if the use of these in vitro 3D models may reduce the number of animals, we believe the in vivo experiments only give the final proof of efficacy and the innocuity of a therapeutic protocol.

## Figures and Tables

**Figure 1 pharmaceutics-15-00806-f001:**
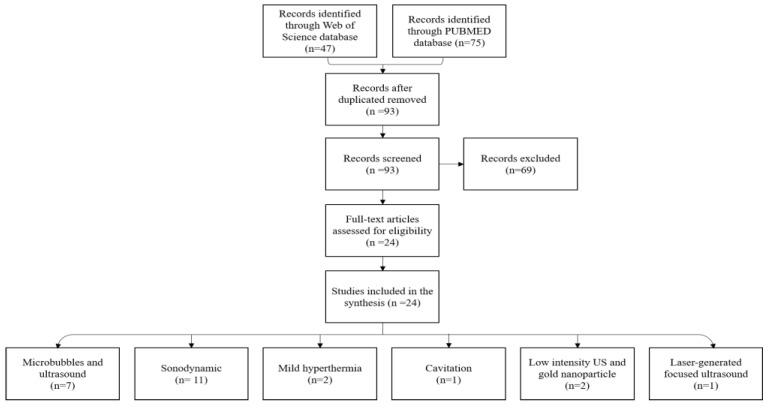
Flowchart of publication selection procedure.

**Figure 2 pharmaceutics-15-00806-f002:**
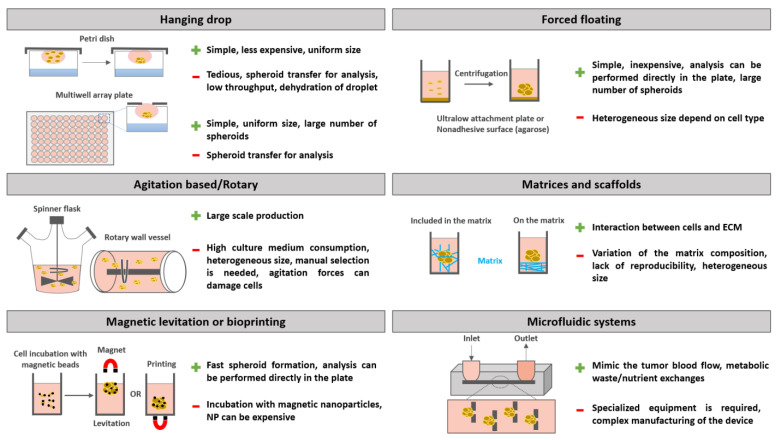
The main methods for spheroid formation. Advantages (**+**) and limitations (**−**).

**Figure 3 pharmaceutics-15-00806-f003:**
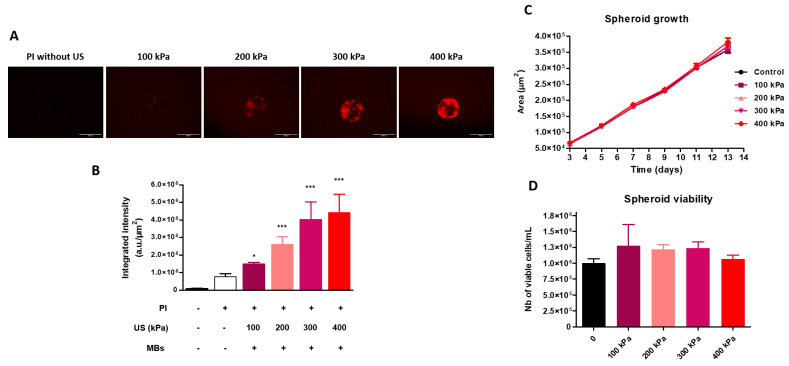
Effect of acoustic pressure on spheroid permeabilization and viability. HCT-116 colon cancer cells were used to generate spheroids in ULA 96-well plates (500 cells/well). At Day 3, when spheroids reached a diameter of 300 µm, the influence of acoustic pressure on cell permeabilization (**A**,**B**), spheroid growth (**C**), and spheroid viability (**D**) was evaluated. To visualize cellular permeabilization, 5 spheroids were transferred to a plastic cuvette under agitation with a magnetic stirrer, and 30 µL of Vevo Micromarker^TM^ MBs (VisualSonics Inc., Toronto, ON, Canada) and 100 µM propidium iodide (PI) were added to McCoy’s medium supplemented with 1% FCS. Spheroids were exposed to 1 MHz sinusoidal US waves with a peak negative pressure of 100 kPa to 400 kPa, burst period of 100 µs, and 40 cycles per pulse for 30 s. The same procedure was applied without PI to evaluate spheroid growth monitored with the EVOS M5000 microscope taking the measurement of the spheroid area with ImageJ software. On Day 13, spheroid viability was assessed using trypan blue staining, and the percentage of viable cells was calculated with Countess automated cell counter. The integrated fluorescence intensity of PI was represented as a function of acoustic pressure. The Mann–Whitney, nonparametric test was used for statistical analysis, and significance was established as * *p* < 0.05 and *** *p* < 0.005 compared to the control condition with PI/without US. Each bar represents the mean ± standard error of the mean (SEM). The scale bar indicates 300 µm.

**Figure 4 pharmaceutics-15-00806-f004:**
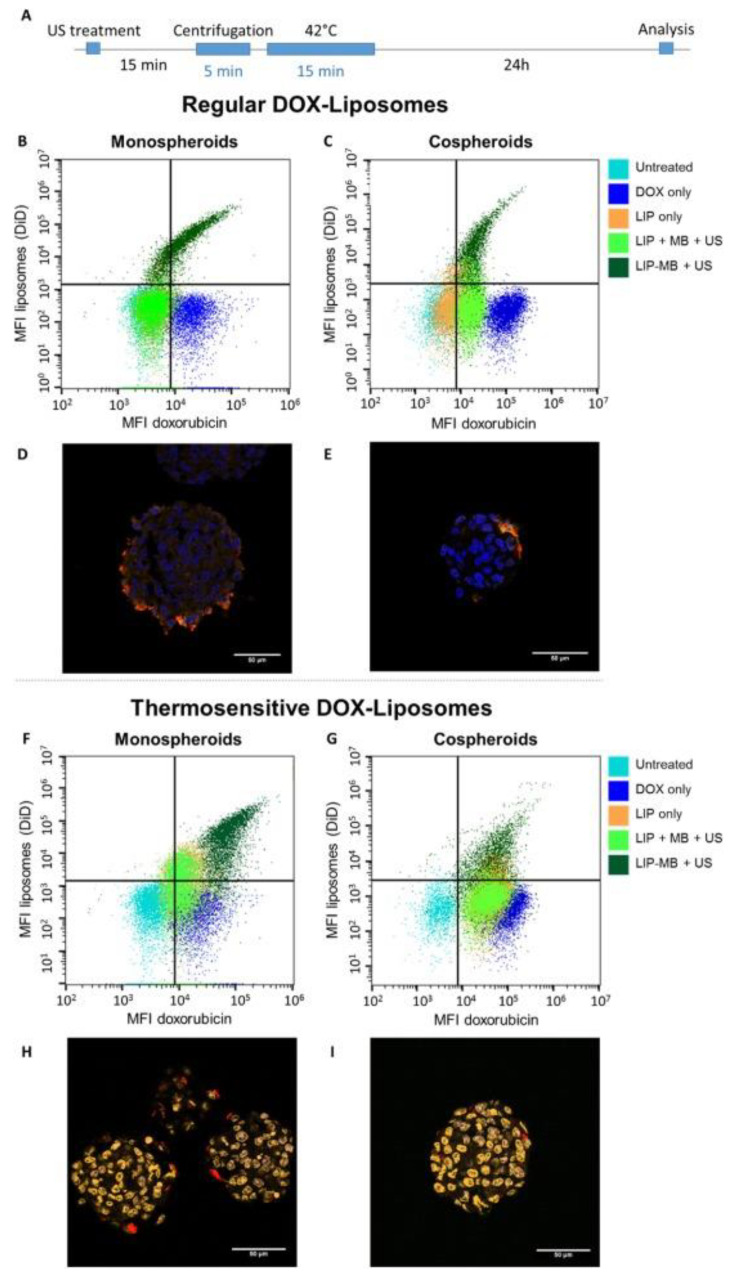
Localization of doxorubicin after sonoprinting and a short heating step. (**A**) Time diagram: 15 min after ultrasound treatment, the spheroids were centrifuged to remove any remaining free doxorubicin or DOX-liposomes. Subsequently, the spheroids were exposed to 42 °C for 15 min. After 24 h, the spheroids were analyzed using flow cytometry and confocal microscopy on cryosections. (**B**,**C**,**F**,**G**) Flow intensity scatter plots of the cellular delivery of doxorubicin (doxorubicin fluorescence, horizontal axis), (**B**–**C**) regular, and (**F**–**G**) thermosensitive liposomes (DiD fluorescence, vertical axis) in (**B**,**F**) monospheroids and (**C**,**G**) cospheroids. DOX = doxorubicin; LIP = DOX-liposomes; MB = microbubbles; US = ultrasound; LIP + MB + US = DOX-liposomes and microbubbles coadministered before ultrasound radiation; LIP-MB + US = DOX-liposomes loaded onto the microbubbles and exposed to ultrasound (i.e., sonoprinting). (**D**,**E**,**H**,**I**) 10 μm cryosections of (**D**,**H**) monospheroids and (**E**,**I**) cospheroids treated with (**D**–**E**) regular or (**H**,**I**) thermosensitive DOX-liposomes coupled onto microbubbles and exposed to ultrasound. Blue = DAPI, orange = doxorubicin, red = liposomes (DiD). The scale bars indicate 50 μm. (For interpretation of the references to color in this figure legend, the reader is referred to the web version of this article). Reprinted from Sonoprinting liposomes on tumor spheroids by microbubbles and ultrasound, Volume 316, S. Roovers, et al., Pages 79–92, Copyright 2019, with permission from Elsevier [70].

**Figure 5 pharmaceutics-15-00806-f005:**
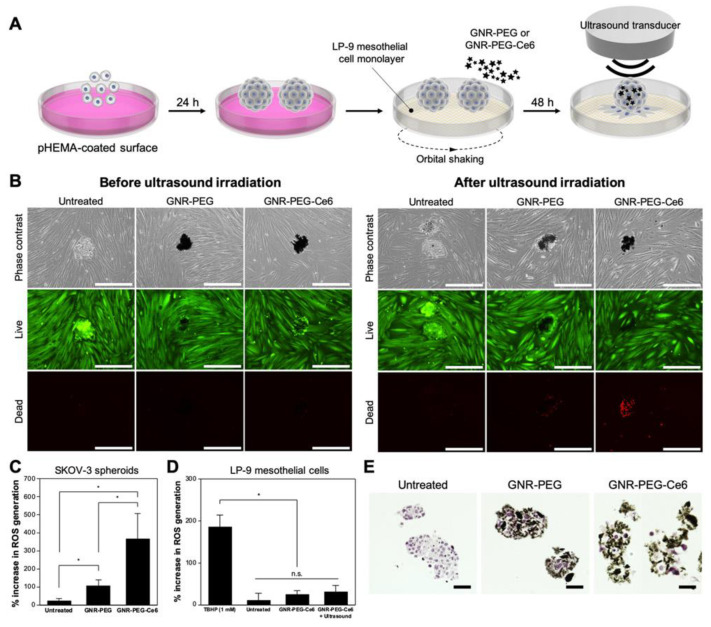
GNR-PEG-Ce6 can kill adhered ovarian cancer spheroids via sonodynamic therapy. (**A**) Schematic of the process for ultrasound irradiation of adhered spheroids. (**B**) Representative images of live (green) and dead (red) cells untreated, GNR-PEG-treated, and GNR-PEG-Ce6-treated SKOV-3 spheroids adhered to the LP-9 mesothelial cell layer before and after ultrasound irradiation. Scale bars indicate 400 µm. ROS generation in (**C**) SKOV-3 spheroids and (**D**) LP-9 mesothelial cells after ultrasound irradiation (* *p* < 0.05; n.s. non-significant). Tert-butyl hydrogen peroxide (TBHP) was used as a positive control in this assay. (**E**) Hematoxylin and eosin-stained (histological) cross-sections of untreated, GNR-PEG-treated, and GNR-PEG-Ce6-treated SKOV-3 spheroids. Scale bars indicate 50 µm. Adapted with permission from Lee et al., copyright 2021, Wiley-VCH [32].

**Figure 6 pharmaceutics-15-00806-f006:**
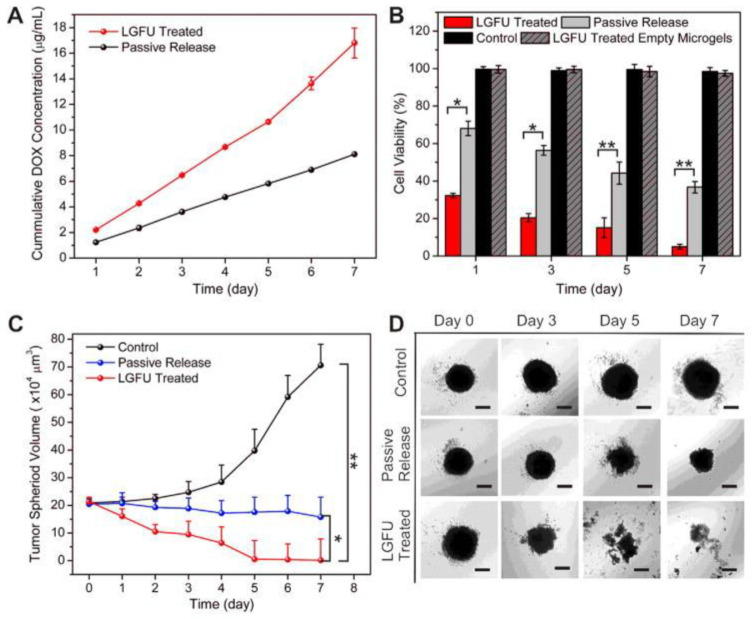
LGFU-induced anticancer effects in vitro. (**A**) The cumulative concentration of released DOX after 30 s treatment with LGFU (18 mJ) and passively released DOX without LGFU treatment. (**B**) Cell viability of HeLa cells treated by solutions from DOX-formulated microgels after 30 s treatment with LGFU (18 mJ) and passive released DOX. (**C**,**D**) Normalized HeLa tumor spheroid sizes and morphologies at day 0, day 3, day 5, and day 7 after treatment with solutions associated with DOX-formulated microgels treated with LGFU, formulations without LGFU treatment (passive release), and PBS (control). Scale bars: 100 μm. Data represents mean ± SD (*n* = 3). * *p* < 0.05 (*t*-test), ** *p* < 0.01 (two-tailed Student’s *t*-test). Reprinted from Spatiotemporal drug delivery using laser-generated-focused ultrasound system, Volume 220 (Pt B), J. Di, et al., Pages 592–599, Copyright 2015, with permission from Elsevier [96].

**Table 1 pharmaceutics-15-00806-t001:** Inclusion and exclusion criteria applied for the selection of publications.

Inclusion Criteria	Exclusion Criteria
Involving US therapeutic on multicellular tumor spheroid for anticancer drug delivery	Without US on spheroid
	Without drug delivery
English	In silico, in vivo
	Review papers, comments, letters
	Other languages

## Data Availability

Other data are available on reasonable request from the corresponding author.

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
