# Peer review of "Tumor Spheroids as Model to Design Acoustically Mediated Drug Therapies: A Review"

_pharmaceutics, 2023, doi:10.3390/pharmaceutics15030806_

Round 1

Reviewer 1 Report

Roy et al. summarized the current studies about acoustically mediated drug therapies on different tumor spheroids models. This topic is interesting. The paper is well written, and the main manuscript is generally clear and concise. The discovery and implications are of scientific interest and I would support publication in Pharmaceuties, but there are some issues that need to be addressed first.

Figure 2.

The front size of contexts within figure is too small to clear identify.

Page 3, line 104-106:

“These ultrasound strategies, including mild hy-104 perthermia, sonodynamic therapy, microbubble-assisted ultrasound and ultrasound on 105 their own, improve the efficacy of anticancer therapies in a safe and noninvasive way”

Please add appropriate citations into this description.

Page 3, line 108-109:

“For a few decades, drug-loaded particles and drug delivery protocols using 108 therapeutic ultrasound have been evaluated in spheroids.”

Please add appropriate citations into this description.

Figure 3.

What is “IP” in the Fig. 3A?

The bottom of Fig. 3D seems missing.

In the section of “Future perspectives and Conclusions”

Please state how to translate to data acquire from 3D tumor models to the in vivo solid tumors.

Table 2

Please add more information of the administrated drug, dye, and particles, such as size and molecular weight.

Author Response

Reviewer 1

Roy et al. summarized the current studies about acoustically mediated drug therapies on different tumor spheroids models. This topic is interesting. The paper is well written, and the main manuscript is generally clear and concise. The discovery and implications are of scientific interest and I would support publication in Pharmaceuties, but there are some issues that need to be addressed first.

We would like to thank the reviewer for his/her comments. We hereby address the points raised by the reviewer. Our answers are given below and highlighted in italic characters. The corrections in the manuscript will be highlighted using the Track Changes feature.

Comment 1 - Figure 2. The front size of contexts within figure is too small to clear identify.

Answer 1 - Corrected

Comment 2 - Page 3, line 104-106: “These ultrasound strategies, including mild hy-104 perthermia, sonodynamic therapy, microbubble-assisted ultrasound and ultrasound on 105 their own, improve the efficacy of anticancer therapies in a safe and noninvasive way”.

Please add appropriate citations into this description.

 Answer 2 – In agreement with the reviewer 1, we added appropriate citations in the introduction section:

  • Deprez J, et al., Opening doors with ultrasound and microbubbles: Beating biological barriers to promote drug delivery. Adv Drug Deliv Rev. 2021 May;172:9-36.
  • Nowak KM et al., Sonodynamic therapy: Rapid progress and new opportunities for non-invasive tumor cell killing with sound. Cancer Lett. 2022 Apr 28;532:215592
  • Entzian K, Aigner A. Drug Delivery by Ultrasound-Responsive Nanocarriers for Cancer Treatment. Pharmaceutics. 2021 Jul 26;13(8):1135.
  • Hijnen N, Langereis S, Grüll H. Magnetic resonance guided high-intensity focused ultrasound for image-guided temperature-induced drug delivery. Adv Drug Deliv Rev. 2014 Jun;72:65-81.

Comment 3 - Page 3, line 108-109: “For a few decades, drug-loaded particles and drug delivery protocols using 108 therapeutic ultrasound have been evaluated in spheroids.”

Please add appropriate citations into this description.

 Answer 3 – We thank the reviewer 1 for this comment. We proposed the following citations:

  • Leenhardt, R. et al., Ultrasound-Induced Cavitation Enhances the Efficacy of Chemotherapy in a 3D Model of Pancreatic Ductal Adenocarcinoma with Its Microenvironment. Sci Rep 2019, 9, 18916.
  • Gao, J. et al., An Ultrasound Responsive Microbubble-Liposome Conjugate for Targeted Irinotecan-Oxaliplatin Treatment of Pancreatic Cancer. Eur J Pharm Biopharm 2020, 157, 233–240.
  • Lee, H.R. et al., Sonosensitizer-Functionalized Graphene Nanoribbons for Adhesion Blocking and Sonodynamic Ablation of Ovarian Cancer Spheroids. Adv Healthc Mater 2021, 10, e2001368
  • Maples, D. et al., Synthesis and Characterisation of Ultrasound Imageable Heat-Sensitive Liposomes for HIFU Therapy. Int. J. Hyperthermia 2015, 31, 674–685

Comment 4 - Figure 3. What is “IP” in the Fig. 3A?

Answer 4 – We apologize for this mistake. We replaced IP by PI, which means Propidium Iodide.

Comment 5 - The bottom of Fig. 3D seems missing.

Answer 5 – Corrected

Comment 6 - In the section of “Future perspectives and Conclusions” - Please state how to translate to data acquire from 3D tumor models to the in vivo solid tumors.

Answer 6 – We thank the reviewer for this relevant comment. We thoroughly reworked the section the section “future perspectives and conclusions” (pages 29 to 31) in the revised manuscript in order to answer to the comments of the 4 reviewers, and thus answer your relevant comment.

Comment 7 - Table 2 - Please add more information of the administrated drug, dye, and particles, such as size and molecular weight

Answer 7 – Corrected.

Reviewer 2 Report

The manuscript by Roy et al. is a nicely written discussion of the role of tumor spheroids in the field of ultrasound drug delivery. First, the authors explain the different culture methods for spheroids, which is very useful for people who are not experts in this field. Then, with a high level of detail, they describe the different applications of tumor spheroids for ultrasound-mediated drug delivery. For this section, they did an extensive search of Pubmed and Web of Science and found all the relevant and recent publications. I have only two minor comments:

- The references in the general description of microbubble-assisted US (section 3.2.1.) and focused ultrasound (3.2.3.) should be more recent and highlight the work of all major groups working in this area.

- Expanding the discussion of 3D vascularized tumor models. In my opinion, the lack of vessels in tumor spheroids is still a major drawback for US-mediated drug delivery applications, as vessels play an important role in transporting drugs and microbubbles to the tumor environment. Much work is being done in the field of vascularized tumor models, it would be nice to connect to this area in this review article.

Author Response

Reviewer 2

The manuscript by Roy et al. is a nicely written discussion of the role of tumor spheroids in the field of ultrasound drug delivery. First, the authors explain the different culture methods for spheroids, which is very useful for people who are not experts in this field. Then, with a high level of detail, they describe the different applications of tumor spheroids for ultrasound-mediated drug delivery. For this section, they did an extensive search of Pubmed and Web of Science and found all the relevant and recent publications. I have only two minor comments:

We would like to thank the reviewer for his/her comments. We hereby address the points raised by the reviewer. Our answers are given below and highlighted in italic characters. The corrections in the manuscript will be highlighted using the Track Changes feature.

Comment 1 - The references in the general description of microbubble-assisted US (section 3.2.1.) and focused ultrasound (3.2.3.) should be more recent and highlight the work of all major groups working in this area.

 Answer 1 – In agreement with the reviewer 2, we added appropriate citations in the sections:

3.2.1.

  • Escoffre JM et al., Delivery of anti-cancer drugs using microbubble-assisted ultrasound in digestive oncology: from preclinical to clinical studies. Expert Opin Drug Deliv. 2022 Apr;19(4):421-433.
  • Zhang N et al., Optimization of microbubble-based DNA vaccination with low-frequency ultrasound for enhanced cancer immunotherapy. Adv Ther (Weinh). 2021 Sep;4(9):2100033.
  • Amate M, et al., The effect of ultrasound pulse length on microbubble cavitation induced antibody accumulation and distribution in a mouse model of breast cancer. Nanotheranostics. 2020 Sep 15;4(4):256-269.
  • Ingram N et al., Ultrasound-triggered therapeutic microbubbles enhance the efficacy of cytotoxic drugs by increasing circulation and tumor drug accumulation and limiting bioavailability and toxicity in normal tissues. Theranostics. 2020 Sep 1;10(24):10973-10992.
  • Sasaki N, et al., Safety Assessment of Ultrasound-Assisted Intravesical Chemotherapy in Normal Dogs: A Pilot Study. Front Pharmacol. 2022 Mar 18;13:837754

3.2.3.

  • Camus M et al., Cavitation-induced release of liposomal chemotherapy in orthotopic murine pancreatic cancer models: A feasibility study. Clin Res Hepatol Gastroenterol. 2019 Nov;43(6):669-681.
  • Di J, et al., Spatiotemporal drug delivery using laser-generated-focused ultrasound system. J Control Release. 2015 Dec 28;220(Pt B):592-9.
  • Lyon PC, et al., Ultrasound Med Biol. Large-Volume Hyperthermia for Safe and Cost-Effective Targeted Drug Delivery Using a Clinical Ultrasound-Guided Focused Ultrasound Device. 2021 Apr;47(4):982-997.

Comment 2 -  Expanding the discussion of 3D vascularized tumor models. In my opinion, the lack of vessels in tumor spheroids is still a major drawback for US-mediated drug delivery applications, as vessels play an important role in transporting drugs and microbubbles to the tumor environment. Much work is being done in the field of vascularized tumor models, it would be nice to connect to this area in this review article.

Answer 2 – We thank the reviewer for this relevant comment. We thoroughly reworked the section “future perspectives and conclusions” (pages 29 to 31) in the revised manuscript in order to answer to the comments of the 4 reviewers, and thus answer your relevant comment.

Reviewer 3 Report

The comprehensive manuscript on tumor spheroids as a model system to study acoustically mediated drug treatments may well serve the researchers who wish to gain overview in the field. The authors clearly declared the inclusion criteria applied in selection of their publication resources. I really appreciate several parts of the manuscript, namely the description of the spheroid cultivation methods including a brief information about their limitations and advantages, as well as the detailed and clearly organized tables (Table 2, 3, and 4).

On the other hand, I don´t fully agree with the authors in one point. 3D spheroids and MCTS do not fully recapitulate the tumor microenvironment, as they are always avascular. In the drug delivery, the extravasation of the drug (delivery system) presents an important aspect. Moreover, MCTS can model the tumor microenvironment to a limited extent only; in real conditions, the complexity of the TME is much greater. Also, the off-target toxicity during the systemic administration is caused by interaction of the treatment with normal cells/tissues and pharmacokinetics. I don´t argue about usefulness of spheroids and MCTS, I would only consider appropriate to mention these limitations more clearly in the Introduction or concluding remarks (e.g., lines 823-827).

It would be important that the authors explain some data to a more details. Specifically, it is comparison of results obtained with 3D spheroids and MCTSs with the data on testing the same treatment in vivo (such as experiments mentioned in lines 585-6 or 677-678).  Relative to the limitations of the 3D model system, these data are extremely important. The use of spheroids and MCT may reduce the number of experimental animals, but the in vivo only gives the final prove.

 Minor point: „trypan blue staining“ is more common than „blue trypan staining“ (e.g., lines 347-8).

Author Response

Reviewer 3

The comprehensive manuscript on tumor spheroids as a model system to study acoustically mediated drug treatments may well serve the researchers who wish to gain overview in the field. The authors clearly declared the inclusion criteria applied in selection of their publication resources. I really appreciate several parts of the manuscript, namely the description of the spheroid cultivation methods including a brief information about their limitations and advantages, as well as the detailed and clearly organized tables (Table 2, 3, and 4).

We would like to thank the reviewer for his/her comments. We hereby address the points raised by the reviewer. Our answers are given below and highlighted in italic characters. The corrections in the manuscript will be highlighted using the Track Changes feature.

Comment 1 - On the other hand, I don´t fully agree with the authors in one point. 3D spheroids and MCTS do not fully recapitulate the tumor microenvironment, as they are always avascular. In the drug delivery, the extravasation of the drug (delivery system) presents an important aspect. Moreover, MCTS can model the tumor microenvironment to a limited extent only; in real conditions, the complexity of the TME is much greater. Also, the off-target toxicity during the systemic administration is caused by interaction of the treatment with normal cells/tissues and pharmacokinetics. I don´t argue about usefulness of spheroids and MCTS, I would only consider appropriate to mention these limitations more clearly in the Introduction or concluding remarks (e.g., lines 823-827). 

Answer 1 – We thank the reviewer for these relevant comments. We qualified our statements in introduction section (pages 1 to 3) and we thoroughly reworked the section “future perspectives and conclusions” (pages 29 to 31) in the revised manuscript in order to answer to the comments of the 4 reviewers, and thus answer your relevant comment.

Comment 2 - It would be important that the authors explain some data to a more details. Specifically, it is comparison of results obtained with 3D spheroids and MCTSs with the data on testing the same treatment in vivo (such as experiments mentioned in lines 585-6 or 677-678).  Relative to the limitations of the 3D model system, these data are extremely important. The use of spheroids and MCT may reduce the number of experimental animals, but the in vivo only gives the final prove.

Answer 2 – In agreement with the reviewer 3, we provided additional details when the authors validated in-vivo their data obtained with 3D spheroids as follows:

585-586 – pages 17 to 18

677-678 – pages 24

As previously mentioned in the answer 1, we nuanced our statements in introduction section (pages 1 to 3) and we thoroughly rephrased the section “future perspectives and conclusions” (pages 29 to 31) in order to answer your relevant comment.

Comment 3 - Minor point: „trypan blue staining“ is more common than „blue trypan staining“ (e.g., lines 347-8).

Answer 3 – Corrected

Reviewer 4 Report

This review introduces the spheroid formation methods in detail and then focuses on acoustically mediated drug therapies. I believe this paper will be of great help to those engaged in the research of nano drug delivery. I recommend publishing this review on Pharmaceutics.

1) I suggest that the descriptions of various methods in Figure 2 are consistent with the order of chapters.

2) Figure 3D shows incompletely.

3) I suggest adding representative pictures to show the experimental ideas or results. This will be more friendly to readers.

Author Response

Reviewer 4

This review introduces the spheroid formation methods in detail and then focuses on acoustically mediated drug therapies. I believe this paper will be of great help to those engaged in the research of nano drug delivery. I recommend publishing this review on Pharmaceutics.

We would like to thank the reviewer for his/her comments. We hereby address the points raised by the reviewer. Our answers are given below and highlighted in italic characters. The corrections in the manuscript will be highlighted using the Track Changes feature.

Comment 1 - I suggest that the descriptions of various methods in Figure 2 are consistent with the order of chapters.

Answer 1 - Corrected

Comment 2 - Figure 3D shows incompletely.

Answer 2 - Corrected

Comment 3 - I suggest adding representative pictures to show the experimental ideas or results. This will be more friendly to readers.

Answer 3 – Corrected